# HPLC and FTIR analysis of phytochemicals and antioxidants of *Aloe vera* exposed to petroleum hydrocarbons and remediation treatments with organic and inorganic amendments

Sarah Alharthi[1,2], Ola A. Abu Ali[1], Amal A. Alyamani[3], Nashi K. Alqahtani[4], Rokayya Sami[5]*, Uguru Hilary[6], Idisi Benjamin Evi[7], Akpokodje Ovie Isaac[8], Haneen H. Mouminah[9], Norah E. Aljohani[10], Mahmoud Helal[11], Salma M. Aljahdali[12], Moayad M. Khashoqji[13], Afnan M. Alnajeebi[14], Hayat A. Alghamdi[15], Sara M. Almutairi[15]

1 Department of Chemistry, College of Science, Taif University, Taif, Saudi Arabia, 2 Research Center of Basic Sciences, Engineering and High Altitude, Taif University, Taif, Saudi Arabia, 3 Department of Biotechnology, College of Science, Taif University, Taif, Saudi Arabia, 4 Department of Food and Nutrition Sciences, College of Agricultural and Food Sciences, King Faisal University, Al-Ahsa, Saudi Arabia, 5 Department of Food Science and Nutrition, College of Sciences, Taif University, Taif, Saudi Arabia, 6 Department of Agricultural Engineering, Delta State University of Science and Technology, Ozoro, Nigeria, 7 Department of environmental management, Delta State University of Science and Technology, Ozoro, Nigeria, 8 Department of Civil and Water Resources Engineering, Delta State University of Science and Technology, Ozoro, Nigeria, 9 Food and Nutrition Department, Faculty of Human Sciences and Design, King Abdulaziz University, Jeddah, Saudi Arabia, 10 Department of Clinical Nutrition, College of Applied Medical Sciences, Taibah University, Medina, Saudi Arabia, 11 Department of Mechanical Engineering, Faculty of Engineering, Taif University, Taif, Saudi Arabia, 12 Department of Biochemistry, Faculty of Science, King Abdulaziz University, Jeddah, Saudi Arabia, 13 Applied College, Taibah University, Madinah, Saudi Arabia, 14 Department of Biological Sciences, College of Science, University of Jeddah, Jeddah, Saudi Arabia, 15 University Medical Clinics, Taif University, Taif, Saudi Arabia

* rokayya.d@tu.edu.sa

## Abstract

Environmental pollution has become a major threat to public safety, and the integrity of bioactive compounds in medical plants. This investigation was conducted to assess the consequence of petroleum hydrocarbons contamination, on the nutritional and medicinal qualities of *Aloe vera* (L.), and to evaluate the efficiency of the various remediation strategies. During the study, soil contaminated with 20% crude oil (petroleum), was treated with five different treatments (T1 to T5), which consist of different concentrations of organic manure, potassium permanganate, and improved seaweed extract manure. All the laboratory tests were conducted by following standard procedures. Specifically, high performance liquid chromatography (HPLC), was used to measure the concentrations of vitamins A and E, as well as the acemannan and aloin, while the Fourier transform infrared spectroscopy (FTIR) analysis, was utilized to evaluate the presence petroleum hydrocarbons inside the extract. The results revealed that, the petroleum contamination and treatment strategies have significant influence, on the extract's phytochemicals and the antioxidant activity behavior. The extract

**Data availability statement:** All relevant data are within the paper and its Supporting Information files.

**Funding:** This research was funded by Taif University, Saudi Arabia.

**Competing interests:** The authors have declared that no competing interests exist.

vitamins A, C and E concentrations varied from 7.68 to 12.47 mg/kg, 1323.67 to 2116 mg/kg, and 54.30 to 73.28 mg/kg, respectively. Additionally, TPC and TFC level of extract varied from 30.15 to 78.50 mg GAE/g, and 9.51 to 38.01 mg QE/g, respectively. The Treatment 4 unit (OM+ISE) showed the best remediation performance, with the highest essential bioactive compounds recovery rates. This affirmed that seaweed extract and organic matters are potential eco-friendly materials, with high efficacy in alleviating the harmful effects, associated with petroleum toxicity on medical plants. This will lead to maximization of the therapeutic benefits of *Aloe vera* plant, grown in petroleum contaminated environment, thus guaranteeing public safety.

## Introduction

Plants have numerous nutritional and pharmaceutical qualities; hence, they play essential roles in supporting human fitness and sustainable medical practices [1]. Medicinal plants often contain abundant bioactive compounds and exhibit adaptability to ecological conditions, though their growth rate is dependent on the species, cropping pattern, habitat influence, as well as other anthropogenic activities [2–4]. Environmental pollution has substantial impacts on the integrity and bioactive compounds of medicinal plants. Chemical compositions and physical characteristics of contaminants can substantially retard plant performance, by inhibiting the biosynthesis of secondary metabolites. Recorded data have that, heavy metals and oily compounds interrupts HMA proteins performance. This interference results in decreased photosynthetic effect, which will consequently lead to a reduction in the phytochemicals' production [5]. Also, pollutants have the ability of interfering with plants DNA performance, leading to severe genetic modification in the plants' metabolite profiles. This subsequently leads to reduction in the plant's pharmacological properties – alkaloids, flavonoids, phenolic acids and other essential minerals concentrations [6–8].

Pollutants linked to petroleum and its derivatives are among the major contributors to environmental pollution globally [9–11]. Petroleum-based pollutants consist of heavy metals (HMs), toxic gases, liquid effluents, volatile compounds such as enzene, toluene, ethylbenzene, polycyclic aromatic hydrocarbons (PAHs), propane and butane. These contaminants have strong ability of compromising the biosynthesis of essential bioactive compounds [12–14]. These disrupt antioxidants production and accumulation of toxic materials within the plant, resulting in a reduction of the therapeutic efficacy of the plant, and severe health challenges to the end users. Decline in the total phenolic content (TPC), minerals, total flavonoid content (TFC), polysaccharides, vitamins, and antioxidants levels in plants, has harmful impact on the herbal drugs produced from them [15]. All these factors significantly revealed that oil contamination, can cause severe disruption on the medical qualities of medical plants. Therefore, it is paramount to alleviate these problems associated with petroleum pollution, through several effective remediation techniques to address oil pollution. The effectiveness hybridized treatments in degrading petroleum products concentration, and subsequently restoring the plant and soil health, have been well-documented [16,17].

A medicinal plant has the capability of providing microbicidal, analgesic, anti-inflammatory, antispasmodic, and antioxidant effects, hence enhancing the development and performance of the human body. Phytochemicals such as phenolics and flavonoid compounds, the major components of medical plants, play vital roles in human defense mechanisms, by enhancing the immune system and inhibiting oxidative stress, thereby protecting the body against disease invasion [18,19]. Likewise, the vitamins and amino acids content in plants, play a great part in improving the body's cellular resilience, protein synthesis, tissue development, and metabolic actions, resulting in an enhanced body immune structure [20]. Moringa (*Moringa oleifera*), turmeric (*Curcuma longa*), *Aloe vera* (L.) Burm.f. (syn. *Aloe barbadensis* Miller), echinacea (*Echinacea purpurea* (L.) Moench), garlic (*Allium sativum),* chamomile (*Matricaria chamomilla*), and neem (*Azadirachta indica*) are some of the plants with high medicinal values. Pharmaceutical attributes of *Aloe vera* plant include anti-inflammatory effect, gastrointestinal support, glucose homeostasis, skin maintenance, and anticancer effect [21,22]. The concentrations of these phytochemicals are correlated with the plant's maturity stage, ecological conditions, and environmental pollution. Weaker concentrations of these vital phytochemicals result in reduced medical outcomes. This can lead to drug failure resulting from the mutations in resistant pathogens [23].

Though the pharmacological effectiveness of *Aloe vera* plant has been well documented [22,24–28], there is still a critical dearth of information on the impacts of petroleum pollution, and consequent remediation strategies, on the phytochemicals and nutritional qualities of *Aloe vera* plant. Therefore, the major novel goals of this research are the assessment of the impact of petroleum contamination and treatment approaches. These will focus on the key pharmaceutical qualities – bioactive compounds of *Aloe vera* leaf extract. Additionally, the efficacy of different soil treatment strategies – organic amendments and inorganic phytoremediation agents – in remediating the harmful effect of petroleum hydrocarbons will be evaluated. Linking *Aloe vera* phytochemical values restoration to soil remediation, this research aims to link the gap between environmental pollution, remediation, and phytotherapy. Specifically, analyses of vitamins, aloin and acemannan will be conducted using the High-Performance Liquid Chromatography (HPLC) technique. Precisely, the Fourier Transform Infrared spectroscopy (FTIR) analysis of the *Aloe vera* extracts will be conducted, to establish a strong relationship between soil contamination levels and *Aloe vera's* therapeutic effectiveness. Moreover, this research will provide valuable information about the ability of inorganic and organic amendment approaches, to restore or modify the therapeutic quality of the *Aloe vera* plant.

## Materials and methods

### Plant material

*Aloe vera* was chosen as the medical plant, primarily due to its high minerals and bioactive compounds contents – aloin, acemannan, phenolic compounds, amino acids, along with vitamins A, C, D, and E. The plant was collected from the research center of the Department of Agricultural Engineering at Delta State University of Science and Technology, Ozoro, Nigeria on 15th July 2024. It was duly identified and authenticated by a crop scientist from the Department of Crop Science, on 22nd July 2024.

### Soil collection and preparation

Three hundred kilograms of topsoil (0–0.4 m depth) was dug from uncontaminated land, while the crude oil used for this experiment was obtained from an oil spill site. This soil depth was chosen since it represents the topsoil, which is rich in microbial activity, good soil texture, high nutrient content, and moderate water retention ability. These conditions will favor plant growth, leading to better phytoremediation performance of the *Aloe vera* plant [29].

The clean soil was impacted with crude oil (petroleum) using specific ratio of 8:1; this means the soil was contaminated by mixing crude oil and soil in an 8:1 mass ratio (soil: crude oil). The soil was thoroughly mixed using a laboratory concrete mixer, to ensure the homogeneity of the mixture of crude oil and soil. This soil contamination level was chosen to have a little novelty from the crude oil volumes used by previous scholars [16,30–32]. This contaminated soil was poured

into perforated drums, and left to stand for two weeks, to allow excess crude oil to seep out, highly volatile petroleum hydrocarbons to evaporate, and the soil to acclimate to the oil contamination.

## Soil amendments (treatments) preparation

Two major soil amendments were used for this research – organic manure (OM) and the seaweed extract-based compost manure (ISE). The organic manure (OM) was formulated by mixing cow dung, poultry waste, kitchen waste, and oil palm empty fruit bunches in a mass ratio of 4:3:2:1. The active aerated composting technique was used to formulate the OM, at a temperature of $32 \pm 5°C$ and a moisture content of $60 \pm 5\%$, for duration of 2 months. The compost was thoroughly turned every 7 days, to facilitate proper aerobic decomposition.

Likewise, the ISE was formulated by blending composted rice husk (prepared through active aerated method), and seaweed extract in a ratio of 3:7. Specifically, in this investigation, *Ascophyllum nodosum* was used as the seaweed source, while the extract was prepared using water-based extraction. The filtrate obtained from the extraction process, was dried (concentrated) in a water bath to obtain approximately 10% stock solution. The analytical-grade potassium permanganate ($KMnO_4$), was produced Thermo Fisher Scientific Inc. America.

## Experimental design and treatment application

This research consists of two major control units, which are the uncontaminated soil and contaminated soil. Specifically, the remediation strategies consist of 5 treatment units (groups), making the experimental program comprising 7 units.
    Control A: Uncontaminated soil
    Control B: Contaminated soil (CS)
    Treatment 1: CS + 2 kg of organic manure (OM)
    Treatment 2: CS + 0.05 kg of Potassium Permanganate ($KMnO_4$)
    Treatment 3: CS + 0.5 kg of improved seaweed extract (ISE)
    Treatment 4: CS + 2 kg OM + 0.2 kg ISE
    Treatment 5: CS + 0.05 kg $KMnO_4$ + 0.5 ISE

Each of the 7 experimental units consists of 7 experimental pots/containers, totaling 49 pots used for this research. These large numbers of experimental pots were used to cushion against the occurrence of plant failure that may occur during the plant growth stage, safeguarding that the research would not be interrupted. For each pot, 15 kg of the contaminated soil was filled into perforated plastic containers (pots), and the appropriate amount of the treatment was incorporated into the soil. Also 15 kg of the uncontaminated soil was filled into a perforated container (Control A). 2 L of borehole water was poured into each unit, and the units were kept in a shady environment for three weeks. This idle period will facilitate the dissipation of the heat generated by the organic manure and inorganic fertilizers, applied to the soil samples as treatments, hence stabilizing the soil's temperature and microbial performance. *Aloe vera* was standardized by planting it in uncontaminated loamy soil for three weeks, during which tap water was provided to the plantlets whenever the soil moisture content fell below 40% (wet basis).

The three-week-old *Aloe vera* plantlets were manually inspected, and only strong, vigorous, and disease-free plantlets were transplanted into the experimental pots. Each experimental unit has seven pots (replicated seven times). Therefore, a total of 49 pots were used for this investigation. One *Aloe vera* seedling was properly transplanted into each pot, during the early evening (5–7 pm) to minimize damage caused by heat stress.

Hybridized treatments were used in this research, to assess the synergistic effects of the treatment agents, to see how their hybridization will enhance nutrients and antioxidants recovery, as the bioactive compounds of these treatments tend to play complementary roles in plant growth. Therefore, the hybridization will enhance nutrients and antioxidants recovery.

After transplanting, the containers were left under the shaded structure, constructed with palm fronds cut from oil palm trees. This will minimize the impact of direct sunlight and precipitations on the plants. Weeding was done manually through

handpicking, also pesticides was not used, as pest control was done through hand picking. This is to prevent the interference of organic or inorganic chemicals in herbicides and pesticides, with the plant's biochemical and DNA structures.

The experiment lasted from September 2024 to January 2025, during which the experimental region experienced approximately 10 hours of sunlight daily. All the experimental pots were subjects to uniform environmental conditions – temperature (24-37°C), relative humidity (81−95%), and wind speed (3.2–12.1 km/h). Since *Aloe vera* is a drought-tolerant plant [33], the plants were irrigated with borehole water when necessary, such as when the soil water level was lower than 30% (wet basis). The moisture content of each experimental pot was measured using a digital soil moisture meter (Model: MC-7828SOIL, having a range of 0–80% and manufactured by Focus Technology Co., Ltd., China). Also, the environmental temperature was measured with a digital thermometer and hygrometer (model: HT-54, temperature range of −9 to 60°C, humidity range of 20–95%) manufactured in India.

## Sample collection and extract preparation

***Aloe vera* leaves.** The leaves were cut from the plant stem with a sharp knife, at the maturity age of 12 weeks after transplanting. These leaves were washed thoroughly with borehole water, sliced, and sun-dried (33±5°C) for 14 days, and the dried leaves were ground using a laboratory grinder. 100 g of the particulate was transferred into a conical flask containing 1 L of ethanol, sealed properly, and kept at room temperature (23-36°C) for 4 days. Then, the fermented product was sifted using a 45 µm sieve to obtain the liquid extract, which remained as the residue on top of the filter paper. Excess solvent was evaporated from the fresh liquid extract, by heating it in a water bath at 50±1°C, dried using a desiccator for 4 hours, and poured into amber-colored bottle and stored at room temperature. The extract yield was calculated through Equation 1 [34].

$$Yield\ (\%) = \frac{M_1}{M_2}$$

(1)

Where: $M_1$ ~ mass of the extract produced, and $M_2$ ~ mass of the materials used for the extract production.

**Soil sample.** The soil content in each pot was emptied into a container, after harvesting the Aloe vera plant for laboratory analysis. The soil was homogenized, kept in a sterilized container covered with aluminum foil, and maintained at 20°C before being used for laboratory analysis.

## Laboratory analysis

**Total petroleum hydrocarbons (TPH) determination.** The TPH concentration in the soil and *Aloe vera* extract samples, were measured in harmony with a standard procedure. The soil sample was sun-dried (30±6°C) for 10 days, then crushed and sieved with a 1.00 mm filter. 150 mL of dichloromethane was poured into 10 g of the soil, which was transferred into a Soxhlet extractor and refluxed for 5 hours. The product was sieved using a 0.45 µm filter, before the GC injection. Additionally, the already produced *Aloe vera* extract was sieved with a 0.45 µm membrane. The gas chromatograph which was fitted with a Flame Ionization Detector (GC-FID) and relevant DB-5 capillary column, was used to measure the TPH concentration in the filtered soil and plant extracts. Remarkably, the GC-FID had these operating parameters – carrier gas (helium) having 1.0 mL/min flow rate, oven temperature varying from 60–280°C, injection temperature of 250°C and volume 1.0 µL. The limits of detection were calculated as 3.3×σ/slope, and the limits of quantification were computed as 10×σ/slope, where σ is the standard deviation. Also, the recovery values varied from 91% to 105%, using a calibration curve ranging from 10 to 2000 mg/kg, with an R² of 0.9985 and a Relative Standard Deviation of 8.1%.

**FTIR assessment.** The FTIR enhances the knowledge of the functional groups and molecular bonds in the extract samples. 2 µL of the extract was carefully positioned onto the ATR (attenuated total reflectance) crystal and then inserted

into the machine. The wavelength was set at the range of 400–4000 cm⁻¹ and utilizing a resolution of 4 cm⁻¹. 30 scans were used for the operation, and the instrument generates a spectral profile, which reveals the peak positions based on the bonds vibrational frequencies [35,36].

**Total phenolic content (TPC) evaluation.** The Folin–Ciocalteu colorimetric screening, which was performed using a UV-Visible Spectrophotometry, was employed to measure the TPC level in each extract specimen. The system uses gallic acid as the benchmark, and 760 nm wavelength. Basically, the Aloe vera extract was prepared by dissolving 10 g of the dried, ground plant leaves in 80% ethanol, and spinning at 4500 rpm for 20 minutes. 0.5 mL of the extract was added to 2.5 mL of diluted Folin–Ciocalteu reagent (prepared by blending the reagent with deionized water at a ratio of 1:10), and the product was incubated at 25°C for 30 minutes in dark condition. The absorption rate was measured using a spectrophotometer at 760 nm, and the TPC calculated through equation 2, which was expressed as mg gallic acid equivalent per gram (mg GAE/g) extract [37].

$$TPC = \frac{mgGAE}{g}$$

(2)

**Total flavonoid content (TFC).** The aluminum chloride (AlCl₃) colorimetric assay was employed to assess the extract's TFC level. This approach employs a UV-Vis Spectrophotometer at 415 nm, and quercetin as the reference standard. The reaction mixture was produced by adding the extract (0.5 mL) to 2.8 mL of distilled water, which was followed by 1.5 mL of ethanol, and 0.1 mL of 10% AlCl₃, as well as 0.1 mL of 1 M CH₃CO₂K. The product obtained was incubated at 27°C for 30 minutes under a dark condition and was followed by the absorbance measurement using the spectrophotometer at 415 nm. The TFC was computed using equation 3, and the answer expressed as mg quercetin equivalent per gram (mg QE/g) extract [38].

$$TFC = \frac{mgQE}{g}$$

(3)

Where: QE is the quercetin equivalent.

**Minerals analysis.** The metals' concentrations in the soil and manure samples were measured via the atomic absorption spectroscopy (AAS) approach. The specimen was sun-dried, ground with a plastic mortar and pestle, and then filtered using a 1 mm filter. One gram of the pulverized sample was added to 10 mL of a mixture of concentrated nitric acid and perchloric acid, combined at the ratio of 4:1, and consequently poured into a digestion tube. This mixture was heated at 100°C until a clear product was achieved, cooled to room temperature (28 ± 4°C), sieved into a container using the Whatman No. 42 filter paper, and diluted to 100 mL by using the distilled water. The metals – copper (Cu), calcium (Ca), sodium (Na), and potassium (K) concentrations – in digested sample were measured, with the aid of the atomic absorption spectroscopy in harmony with approved procedures. The absorbance wavelengths used were 324.8 nm, 422.7 nm, 589.0 nm, and 766.5 nm for copper, calcium, sodium, and potassium, respectively.

Furthermore, the soil and manure specimens; nitrogen (N) and phosphorus (P) concentrations were determined, by utilizing the Kjeldahl and Bray techniques, respectively. To measure the N level, 1 g of the filtered sample was digested (100°C) with 15 mL of concentrated sulfuric acid inside a Kjeldahl flask. The product was cooled, sieved, diluted with distilled water to 100 mL, added a few drops of NaOH, and the ammonia released was distilled into a boric acid solution. Then, the total nitrogen was calculated using Equation 4. Similarly, to measure the phosphorus level, 5 g of the crushed sample was mixed with 50 mL of extractant, which was shaken for 10 minutes, and sieved using a 45 µm membrane. 5 mL of this prepared aliquot was added to a molybdate reagent, and the phosphorus concentration was measured using a spectrophotometer at 880 nm

$$N = \frac{(V_{sample} - V_{blank}) \times N_{acid} \times 14.01}{sample\ weight} \times 100$$

(4)

## High-Performance Liquid Chromatography (HPLC) analysis

**Vitamins A and E determination.** The vitamins A and E content of the extract were determined through the HPLC approach. 20 mL was mixed with 80 mL of n-hexane, and centrifuge at 7000×g for 7 min. The upper stratum containing mainly the vitamins was separated, evaporated utilizing the rotary evaporator, sifted utilizing a 0.22 μm mesh, and the filtered fluid was used for the HPLC analysis (PAN et al., 2020). Specifically, the operating parameters used to determine the extract vitamins A and E quantities are given in Table 1, and the results were interpreted by the instrument's software, which used the information obtained from the detector, and then displayed as shown in Table 2, with the details in Supplementary S1 Fig in S1 File.

**Aloin determination.** The aloin concentration in the *Aloe vera* was also measured using the HPLC method. 30 mL of the sample was sieved with 0.45 μm syringe sieve, before the HPLC analysis. 20 μL of the aliquot was introduced into the HPLC machine, and the chromatography procedure was done with a C18 reversed-phase column (250×4.6 mm, 5 μm) at a wavelength of 270 nm, basically utilizing the parameters stated in Table 1. The quantification was done using standardization with a 5 μg/mL standard aloin solution, and the unit of result was mg/g.

**Acemannan determination.** The specimens' acemannan level was measured using the HPLC system, which was equipped with a Refractive Index Detector (RID). The extract was solubilized using distilled water, which was heated to 40°C for 60 minutes, before it was sifted using a 0.45 μm microfilter. 30 μL of the prepared solution was introduced into the machine, and the chromatographic operation was performed using $NH_2$ column and distilled water as the mobile phase. The parameters utilized by the machine to achieve the goal are given in Table 1. The unit of the acemannan concentration was mg/g dry weight (DW).

**Vitamin C determination.** The vitamin C level in the extract was quantified using the UV–Vis Spectrophotometry approach. 10 mL of extract and 80 mL of distilled water were mixed together and sieved using Whatman No. 1 filter paper. The resulting product was diluted to 100 mL by using 0.1 M tetraoxosulphate (VI) acid. Then, the vitamin C concentration in the prepared sample was determined with a UV-visible spectrophotometer at 265 nm [39].

**Amino acids determination.** The amino acids content in the soil, manure, and extract was analyzed, by using the automated amino acid analyzer (model LA8080 Hitachi, manufactured in Japan). The machine has column dimensions of 4.6 mm by 60 mm, a wavelength of 570 nm, and accepts injection volume up to 100 μL. The soil and manure specimens

**Table 1. HPLC operating parameters.**

| Parameter | Aloin | Acemannan | Vitamin A | Vitamin E |
|---|---|---|---|---|
| **Detection Type** | UV – visible | Refractive Index | UV – visible | UV – visible |
| **Injection Volume (μL)** | 20 | 30 | 20 | 20 |
| **Column Type** | C18 RP | $NH_2$ | C18 RP | C18 RP |
| **Mobile Phase** | Water: Acetonitrile (8:2) | Distil Water | Methanol: Water (95:5) | Methanol |
| **CT (ºC)** | 29 | 30 | 30 | 25 |
| **FT (L/m)** | 0.001 | 0.001 | 0.001 | 0.001 |
| **WL (nm)** | 270 | 210 | 330 | 290 |
| **Run Time (min)** | 20 | 25 | 12 | 11 |
| **Retention Time (min)** | 10 | 20 | | 10 |

WL = Wavelength, CT – Column temperature, RP = Reverse phase, FT – Flow rate.

**Table 2. The vitamins A and B Chromatogram and numbers.**

| Peak No. | RT (min) | PA (mAU·s) | PH (mAU) | Area (%) | Compound Identified |
|---|---|---|---|---|---|
| **Chromatogram of Vitamin A** | | | | | |
| 1 | 1.98 | 350,432 | 11,245 | 2.75 | Unknown impurity |
| 2 | 3.54 | 1,254,675 | 38,590 | 9.84 | All-trans-retinol |
| 3 | 4.92 | 11,040,821 | 135,425 | 86.47 | Retinyl acetate |
| 4 | 6.33 | 214,080 | 4,813 | 0.94 | Trace component |
| **Chromatogram of Vitamin E** | | | | | |
| 1 | 2.23 | 289,500 | 8,912 | 2.61 | Unknown |
| 2 | 3.78 | 1,065,740 | 25,875 | 9.62 | δ-Tocopherol |
| 3 | 5.41 | 3,842,655 | 79,845 | 35.52 | γ-Tocopherol |
| 4 | 6.98 | 5,573,240 | 91,234 | 51.80 | α-Tocopherol |
| 5 | 8.11 | 182,540 | 4,675 | 0.45 | Unknown trace |

RT – Retention time, PA – Peak Area, PH – Peak Height.

were initially air-dried, crushed, and sifted using a 1 mm sieve. Specifically, the amino acids in the soil, manure, and *Aloe vera* extract were extracted by adding 10 g of the extract to 20 mL of 6 N HCl, which contains 0.1% phenol at $110 \pm 2$°C for 24 h, inside a hydrolysis tube. The hydrolysate produced was sieved, and evaporated using the water bath set at 50°C to remove the excess acid. 2 mL of 0.2 M citrate was added to the residue, which was vortexed, sifted using a 0.45-micron filter, and stored in vials. Thereafter, 40 μL of the aliquot was injected into the analyzer, with the column temperature kept at 60°C, and the wavelength at 570 nm. The total amino acids were identified and expressed as mg/100 mL [40].

**Antioxidant activity.** The 2,2-Diphenyl-1-picrylhydrazyl (DPPH) approach, was used to determine the extract antioxidant activity level. A 0.1 mM DPPH solution was made with methanol (2 mL DPPH + 1 mL extract). This resulting mixture was agitated vigorously, subjected top incubation (30°C) for 30 minutes, after which the absorbance level was read at 517 nm using the UV – visible spectrophotometer (model – Evolution 220, produced by Thermo Fisher Scientific Inc., America). The blank used for the calculation was only the DPPH and the methanol, and the inhibition rate was computed using Equation 5 [41].

$$\%Inhibition = \frac{Abs_{blank} - Abs_{spe}}{Abs_{blank}}$$

(5)

Where: $Abs_{blank}$ ~blank absorbance and $Abs_{spe}$ ~ specimen absorbance reading

**Recovery rate.** The recovery rates of the treatments were calculated, in related to the Control A results, as presented in Equation 6.

$$Recovery\ rate\ (\%) = \frac{specific\ treatment\ value}{Control\ treatment\ value} \times 100$$

(6)

## Statistical analysis

The results will be analyzed by utilizing the one-way analysis of variance (ANOVA), through the aid of the Statistical Package for the Social Sciences (SPSS) version 20.0. This is to establish the impact of the treatments on the yield and phytochemicals of the *Aloe vera* extract. Also, Duncan's Multiple Range Test (DMRT) which is a post-hoc test, was be used to compare and separate the average values at significance level of 5% ($p \leq 0.05$). Each test was conducted in triplicate

## Results and discussion

### Preliminary investigation into the soil and manure properties

The results of the essential components of the uncontaminated soil, contaminated soil, organic manure and seaweed extract-based compost manure are presented in Table 3. The geotechnical assessment of the soil shows that it belongs to the sandy loam category, and its water holding capacity is 24.93%. Hence, it can support *Aloe vera* growth and productivity. Succulent plants like *Aloe vera* thrive in sandy loam, with moderate a water holding capacity of about 30%, which promotes vigorous root growth [42]. As shown in Table 3, the soil TPH level increased from 23.88 to 2,419.48 mg/kg after the crude oil contamination. It was noted that crude oil contamination caused substantial alteration in the soil biochemical properties, leading to reduction in the soil organic matter content, amino acid concentration and water holding capacity. Petroleum has the ability to block the soil pores, and form impermeable layers around the soil grains, resulting in poor water holding capacity and nutrient availability. This causes serious problems for *Aloe vera* performance, and subsequently the concentration of its extract pharmacologically active compounds [17,32]. The elevated C:N ratio of the contaminated soil relative to the uncontaminated soil further confirmed the adverse impact of petroleum contamination on the soil, since petroleum is rich in carbon compounds.

Additionally, the results show that both OM and ISE, contain considerable amounts of amino acids. Amino acids are essential components of the soil that enhance plant productivity. These acids act as natural chelators, hence increasing nutrients bioavailability and assimilation, as well as boosting the plant resistance to abiotic environmental challenges. This promotes high plant yield be better nutritional qualities. Therefore, the ISE and OM, when applied as natural additives, will promote microbial activity and plant survivability, thereby increasing bioremediation of contamination soil, and

**Table 3. The soil and amendments physiochemical and biological properties.**

|  | Uncontaminated soil | Contaminated soil | Organic manure | ISE |
|---|---|---|---|---|
| **Physical properties** | | | | |
| Sand (%) | 74.19 | – | – | – |
| Silt (%) | 17.84 | – | – | – |
| Clay (%) | 7.97 | – | – | – |
| SC | Sandy Loam | | | |
| **Chemical analysis** | | | | |
| TPH (mg/kg) | $23.88^a \pm 0.31$ | $2419.48^b \pm 24.29$ | – | – |
| pH | $7.03^a \pm 0.03$ | $7.92^b \pm 0.03$ | $8.49^d \pm 0.04$ | $8.16^c \pm 0.02$ |
| **Total nitrogen (mg/kg)** | $12.19^a \pm 0.04$ | $7.26^a \pm 0.02$ | $13,923.13^c \pm 100$ | $10,227.04^b \pm 44.57$ |
| P (mg/kg) | $10.37^a \pm 0.02$ | $8.93^a \pm 0.02$ | $2,844.97^c \pm 50.27$ | $1,642.55^b \pm 27.08$ |
| Cu (mg/kg) | $7.26^b \pm 0.01$ | $9.74^d \pm 0.04$ | $8.45^c \pm 0.05$ | $3.29^a \pm 0.03$ |
| Ca (mg/kg) | $2,261.31^a \pm 20.00$ | $2,093.66^b \pm 40.0$ | $18,381.55^c \pm 9.05$ | $20,773.87^d \pm 105.3$ |
| Na (mg/kg) | $359.82^c \pm 8.14$ | $411.38^d \pm 3.00$ | $85.49^a \pm 1.59$ | $135.42^b \pm 2.36$ |
| K (mg/kg) | $187.17^a \pm 4.75$ | $174.99^a \pm 7.57$ | $8,494.14^c \pm 91.37$ | $6,495.35^b \pm 39.48$ |
| ORM (%) | $6.15^a \pm 0.11$ | $5.24^a \pm 0.08$ | $47.09^c \pm 1.57$ | $34.36^b \pm 0.09$ |
| WHC (%) | $24.93^b \pm 0.03$ | $16.31^a \pm 0.14$ | $55.26^c \pm 1.37$ | $51.49^c \pm 2.33$ |
| AC (mg/kg) | $63.51^b \pm 1.98$ | $38.27^a \pm 4.25$ | $2,637.82^c \pm 30.38$ | $36,772.63^d \pm 113.12$ |
| C:N ratio | 62:1 | 16:1 | 11:1 | 13:1 |
| **Moisture content (%)** | $16.37 \pm 0.95$ | $47.92 \pm 0.88$ | $8.33 \pm 0.04$ | $21.48 \pm 1.15$ |

± - mean and standard deviation, SC – Soil classification, WHC – Water holding capacity, Cu – copper, Ca – calcium, Na – sodium, K – potassium, P – Phosphorus, ORM – Organic matter, ISE – Seaweed extract-based compost manure, AC – Total Amino acids. Specifically, within the same parameter, columns having the same alphabet in superscript indicate that the means are significantly similar (p ≤ 0.05) according to DMRT.

assisting in essential nutrients recovery [43]. The appreciable organic matter content of the organic manure and seaweed manure will enhance microbial performance and plant productivity, thereby promoting phytoremediation of the petroleum-contaminated soil. Organic matter provides necessary nutrients and water for soil microorganisms, which facilitate enzymatic functionality and degradation of toxic hydrocarbons. Moreover, organic matter is rich in macronutrients and helps improve the soil macrostructural pattern and water retention ability, leading to better *Aloe vera* growth, effective phytoremediation, and the promotion of ecological equilibrium [44].

### *Aloe vera* extract's yield and TPH level

The results – including the crude extracts yield, the extract TPH value and soil TPH concentrations, are presented in Table 4. The extract yields of control A, control B, and experimental Treatments 1–5 were 21.21, 9.66, 16.35, 13.75, 14.95, 18.09 and 15.90%, respectively. The results further revealed that the petroleum contamination and the remediation program have significant effects on the *Aloe vera* extract yield. Typically, it was noted that plants in Control B displayed physical appearance of TPH toxicity, along with stunted growth, chlorosis, and wilting, and some of the plants failed to thrive. In spite of some deviations, the combined treatment of organic manure and ISE gave the best remediation option, with an extraction yield of 18.09%. This is an indication that organic materials and biostimulant (seaweed) aid metabolic activities, leading to increased plant extract yield. Organic matter and natural amendments tend to increase the soil fertility levels, enhance the plant's ability to resist both the biotic and abiotic stresses; hence maximizing extract's yield and bioactive compound production [45]. The *Aloe vera* extract yields in this research, primarily among the remediated experimental units, are like the values stated by Mehmood [46]. This finding of the decline in the plant's yield following petroleum contamination, and the subsequent increase in its metabolism after remediation is augmented by previous reports [16,47].

Also, the soil TPH and extract TPH results (Table 4) revealed that the treatments have a significant influence on the TPH concentration, both in the soil and extract ($p \leq 0.05$). The mean THP levels in the soil and plant extract after the experimental period ranged from 20.59 to 2022.25 mg/kg, and 1.11 to 360.95 mg/kg, respectively. This is aligned to Uguru observation [12], which documented that petroleum pollution caused a drastic upsurge in the quantity of PHs in the soil. Generally, the untreated unit exhibited the highest soil TPH concentration both in the soil and in the extract, indicating the positive effects of the treatment agents, in degrading the petroleum hydrocarbons in the soil. Also, the results revealed organic matters-based treatments (T1, T3 and T4) gave better results, compared to treatments based on inorganic materials (T2 and T5). Apart from fixation of nutrients (mainly potassium) in the soil, which will enhance phytoremediation through the improved plant growth, potassium permanganate could form reactive species, which can oxidize the complex petroleum molecules [48,49]. Natural and synthetic remediating materials help to improve crude oil biodegradability, support microbial activity, and enhance plant's essential nutrients recovery. These actions play vital roles in degrading

**Table 4. The TPH concentration and *Aloe vera* extract yield.**

| Source | Extract yield (%) | Soil TPH (mg/kg) | Extract TPH (mg/kg) |
|---|---|---|---|
| Control A | 21.21[g] ± 0.90 | 20.59[a] ± 0.68 | 1.11[a] ± 0.29 |
| Control B | 9.66[a] ± 1.13 | 2022.25[g] ± 14.18 | 360.95[g] ± 6.14 |
| T1 | 16.35[e] ± 0.66 | 919.40[d] ± 16.72 | 183.69[d] ± 6.30 |
| T2 | 13.75[b] ± 0.47 | 1124.06[f] ± 5.13 | 225.17[f] ± 5.22 |
| T3 | 14.95[c] ± 0.42 | 1014.47[e] ± 11.47 | 201.70[e] ± 3.56 |
| T4 | 18.09[f] ± 0.61 | 460.42[b] ± 24.91 | 111.65[b] ± 7.68 |
| T5 | 15.90[d] ± 0.48 | 626.07[c] ± 3.13 | 153.69[c] ± 1.84 |

TPH – total petroleum hydrocarbons; within the same column, rows having the same alphabet indicate that the means are significantly similar ($p \leq 0.05$) according to DMRT.

petroleum hydrocarbons, thus lowering their concentration in the host medium, and immensely contributing to environmental and public health restoration [50]. Interestingly, this research has some noteworthy long-term effects, especially the treatments with larger proportions of biomaterials, as they will improve both the geotechnical and biochemical properties of soil. This will enhance public health safety and environmental conservation.

## Nutritional concentration and recovery rates

The results of the *Aloe vera* extract, including vitamins A, C, and E, and amino acids, in addition to their recovery rates, are presented in Table 5 and Figs 1 to 2. Typically, under the different treatments, the vitamins A, C and E content of the extracts varied from 7.68–12.47 mg/kg, 1323.67–2116 mg/kg and 54.30–73.28 mg/kg, respectively. Generally, the ranking of vitamins in the extracts followed this pattern: vitamin C > vitamin E > vitamin B, indicating that vitamin C recorded the highest vitamin concentration regardless of the treatment applied. Likewise, the mean amino acids level in the extract samples ranged from 27.05 to 98.35 mg/100 mL. The findings highlighted that petroleum contamination caused a substantial decrease in the concentration of the vitamins investigated. This depicted that vitamins are prone to oxidative stress and are susceptibility to environmental stressors, including petroleum hydrocarbons [53]. The significant decline in *Aloe vera* essential vitamins, triggered by the oil contamination, will hinder the plant's ability to perform its critical clinical benefits, such as immune system improvement, tissue development, and cellular resistance [22]. Petroleum toxicity leads to damage of the plant root membrane damage, resulting in poor nutrient absorption and disruption of the plant Metabolism system. Subsequently, these cause significant decline in the biosynthesis of vital nutrients, such as vitamin and other secondary metabolites production [54,55].

Specifically, the treatment materials substantially increased vitamins production, in the plant body system, resulting in high recovery rates among the vitamins investigated. The results highlight that T4 had the best vitamin recovery rate, regardless of the type of vitamin, and this can be linked to the presence of seaweed extract in the treatment plan. Explicitly, the vitamins A, C, and E concentrations, in the *Aloe vera* planted in contaminated soil increased by 48.58, 54.15, and 39.89%, respectively, after T4 application. According to the study's results, the T4 amendment was able to restore the vitamin levels in the extract almost to the concentration, recorded in the extract obtained from *Aloe vera* planted in uncontaminated soil. The maximum recovery rate recorded in T4 does not mean that this treatment contains the higher amounts of active phytochemicals; there are instances of antagonistic interactions between the hybridized treatments, consequently leading to poor plant performance [52]. Seaweed extract and organic manure are rich in essential minerals, cytokinins, phenolic acids, antioxidants, microbes, and organic matter. These compounds are effective in degrading hydrocarbons in the soil, as well as enhancing nutrient availability and plant metabolic enzymatic reactions. This leads to an increase in plant functionality and vitamin synthesis [51,52].

**Table 5. The extract vitamins and amino acids levels.**

| Source | Vitamin A (mg/kg) | Vitamin C (mg/kg) | Vitamin E (mg/kg) | Amino acids (mg/100 mL) |
|---|---|---|---|---|
| **Control A** | 15.66[g] ± 0.41 | 2554.33[g] ± 44.88 | 88.82[g] ± 1.02 | 98.35[g] ± 2.93 |
| **Control B** | 6.41[a] ± 0.40 | 970.00[a] ± 7.55 | 44.04[a] ± 1.59 | 27.05[a] ± 2.70 |
| **T1** | 11.17[e] ± 0.10 | 1966.67[e] ± 8.50 | 69.89[e] ± 1.26 | 66.35[d] ± 2.06 |
| **T2** | 7.68[b] ± 0.18 | 1323.67[b] ± 22.55 | 54.30[b] ± 1.12 | 51.70[b] ± 2.31 |
| **T3** | 8.57[c] ± 0.14 | 1554.67[c] ± 9.71 | 59.71[c] ± 3.04 | 59.44[c] ± 1.68 |
| **T4** | 12.47[f] ± 0.46 | 2116.00[f] ± 6.56 | 73.28[f] ± 1.07 | 91.35[f] ± 3.33 |
| **T5** | 10.25[d] ± 0.33 | 1674.00[d] ± 60.51 | 64.82[d] ± 0.71 | 73.30[e] ± 3.15 |

Rows sharing the same alphabet in the same column specify that these means are not significantly diffenrt at p ≤ 0.05 according to DMRT.

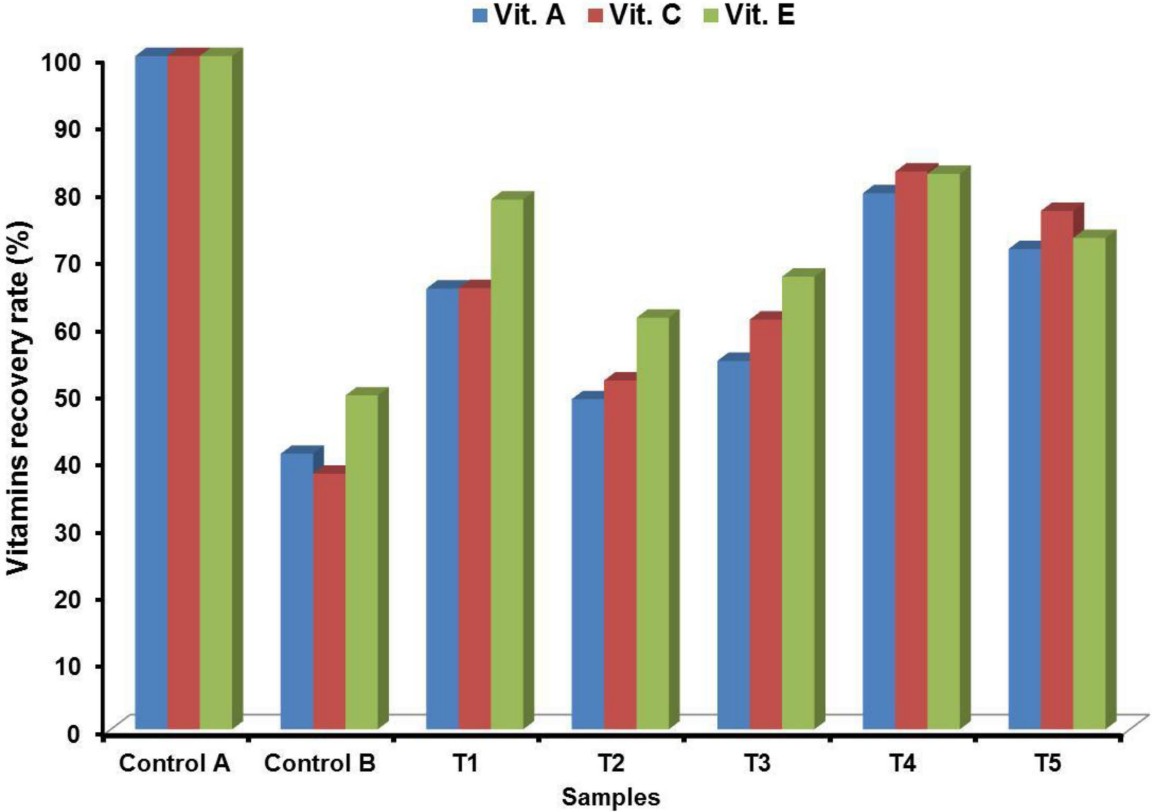

**Fig 1. The vitamins' recovery rate.**

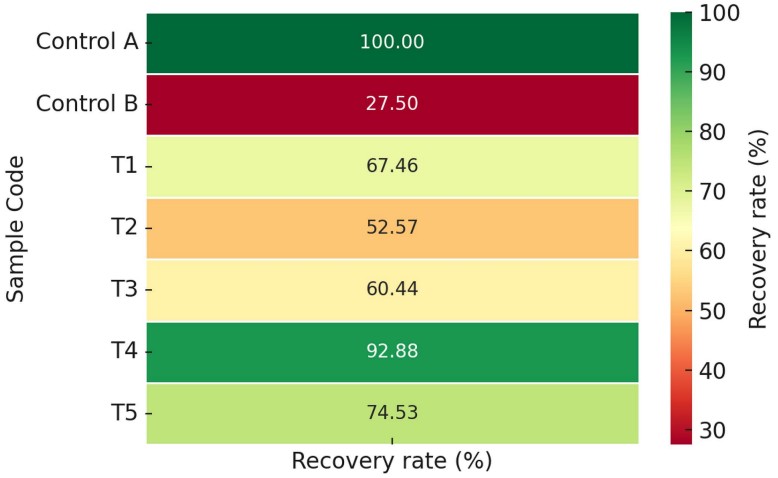

**Fig 2. Heatmap of the amino acid recovery rate.**

Furthermore, it was detected in the results that the remediation techniques led to significant recovery in the concentrations of vitamins A, C and E (Fig 1). Control A had the maximum recovery rate (100%), while Control B recorded the lowest recovery rate. Generally, among the treatments, the Treatment 4 recorded the highest recovery values, displaying consistent high recovery rates across the three vitamins (A, C, and E). This situation can be correlated the high organic materials content in these treatments. Linking these outcomes to the TPH degradation results (Table 4), it gives an indication that the treatment strategies effectively detoxified the contaminated soil and further restored the plant's nutritional profile. Basically, vitamin C recorded the maximum recovery rate, which is an indication that the organic materials and biostimulants, facilitated massive vitamin C production in the plant. These findings buttressed Rostaei [4] report, which stated that organic manure and biostimulants help optimize biochemical routes, and boost secondary metabolites production, resulting in an increment in antioxidants synthesis in plants. Organic materials enhance microbial performance under favorable soil conditions, leading to soil detoxification and the formation of essential bioactive compounds [56]. Vitamins A, C and E have numerous therapeutic advantages, and with specific prospective of disease prevention, such as anti-inflammatory effects, collagen production, inhibition of neurodegenerative diseases, enhances vision health, and improvement of body immunity.

Furthermore, the results presented in Fig 2 further established that the amino acids recovery rate in Controls A and B, as well as Treatments 1–5 experimental units. Specifically, Control A exhibited the highest recovery rate (100%), followed by T4, which recorded an amino acid recovery rate of 92.88%. On the contrary, Control B recorded the lowest recovery rate of 27.50%, followed by T2, which had a recovery rate of 52.57%. The heatmap visualization has affirmed that the remediating agents, immensely help to recover the plant's amino acid content, which was higher in the combined treatments. The superior performance of hybridized treatments, regarding petroleum remediation and amino acid recovery rate, could be correlated to the greater concentrations of plants' nutrients and remediating bioactive compounds (antioxidants and amino acids, as shown in Tables 3 and 6. Also, the synergy among these large proportions of bioactive compounds in the combined treatments will enhance treatment effectiveness, resulting in better decontamination and enhanced plant metabolic performance [57]. Additionally, the high amino acids and essential nutrients content, of the seaweed-based manure help to boost microbial activity and enhance plant growth. Specifically, the amino acids increase the plant's resistance to environmental stress and metabolic efficiency, resulting in an increase in amino acids formation in the plant body [43,58].

This research's findings have highlighted that organic-based treatment has proven better performance in restoring the plant bioactive compounds, DPPH scavenging activity, and degrading the petroleum hydrocarbons levels in the soil. This affirmed previous scholars' reports that organic materials, including biostimulants, have a higher prospective for remediating contaminated soil and restoring the soil's geotechnical properties [59–61]. This bioremediation can be carried

**Table 6. Major bioactive compounds and antioxidant actions of the *Aloe vera* extract.**

| Source | Aloin (mg/g) | Acemannan (mg/g) | DPPH (% Inhibition) | TPC (mg GAE/g) | TFC (mg QE/g) |
|---|---|---|---|---|---|
| Control A | 4.70±0.07 | 5.72[g]±0.04 | 70.35[f]±2.83 | 78.50[g]±1.35 | 38.01[g]±1.76 |
| Control B | 2.23±0.03 | 2.70[a]±0.02 | 21.62[a]±1.81 | 30.15[a]±0.60 | 9.51[a]±0.66 |
| T1 | 3.58±0.04 | 4.42[d]±0.04 | 41.72[c]±1.61 | 57.38[d]±1.45 | 21.67[d]±1.57 |
| T2 | 2.94±0.03 | 3.27[b]±0.06 | 28.48[b]±0.85 | 48.29[b]±0.55 | 14.73[b]±1.36 |
| T3 | 3.33±0.03 | 4.16[c]±0.08 | 39.22[c]±1.93 | 51.81[c]±1.32 | 17.63[c]±0.74 |
| T4 | 4.17±0.06 | 5.19[f]±0.04 | 60.99[e]±1.31 | 69.96[f]±2.31 | 31.94[f]±2.25 |
| T5 | 3.85±0.04 | 4.59[e]±0.04 | 49.35[d]±1.93 | 60.43[e]±1.37 | 25.36[e]±1.63 |

TFC – total flavonoid content; TPC – total phenolic content; within the same column, the rows that shared similar alphabet indicate that the separated means did not exhibit significantly difference according to DMRT (p ≤ 0.05).

out without negatively impacting the soil's biochemical properties. Additionally, the organic compounds in the manure immensely help to facilitate the activities of the hydrocarbon-utilizing microorganisms in the soil, leading to the rapid breakdown of the petroleum present in the contaminated soil [43]. This leads to the rapid recovery of the vitamins and amino acids of the *Aloe vera* plant; thereby increasing the extract's nutritional and antioxidant values.

Heatmap legend interpretation: green color signifies higher amino acid recovery rates, while the red and yellow areas indicate lower recovery rates

## Phytochemical composition and antioxidant properties

The results of the concentrations of aloin and acemannan in the extract samples, as well as the DPPH scavenging activity, total phenolic content (TPC), and total flavonoid content (TFC), are presented in Table 6. Statistically, the treatments intervention had significant influence on antioxidants level and bioactive compounds of the *Aloe vera* extracts ($p \leq 0.05$). The aloin levels in the different extracts varied from 2.23 to 4.70 mg/g, the acemannan concentrations ranged from 2.70 to 5.72 mg/g, the DPPH scavenging activity values varied between 21.62 and 70.35%. Notably, the TPC and TFC levels in the extract declined sharply (61.58% and 74.99% respectively) after petroleum contamination. These parameters (TPC and TFC) increased gradually after the soil amendments, with T4 sample attaining the maximum TPC and TFC recovery rates (89.22% and 84.03%, respectively) among the five treatment units. Among the treatments, it was observed that T2 exhibited the least antioxidant activity, recording DPPH inhibition value of 21.62%. The results also depicted that Treatment 2 had the lowest TPC and TFC values of 48.29 mg GAE/g and 14.73 mg QE/g, respectively. The poor antioxidant activity recovery effectiveness, of this treatment can be linked to the inability, of potassium permanganate to retard oxidative stress, and enhance phytochemical biosynthesis. This is mainly due to fewer essential bioactive compounds, and organic matter contained in potassium permanganate (Treatment 2).

As shown by the results, petroleum contamination adversely affects the extracts antioxidants properties, as the extracts' aloin, acemannan and antioxidant activity levels declined by 52, 53, and 70%, respectively (differences between Control A and Control B). Similarly, the treatment strategies were observed to play significant roles in improving the levels of the aloin, acemannan and antioxidant activity of the *Aloe vera* leaves extract. Regrettably, the decline in the concentrations of the bioactive compounds (acemannan, aloin, TFC and TPC), will have substantial effect on the therapeutic effectiveness of the *Aloe vera* extract [21]. This is confirmed by the remarkably lower DPPH level recorded in the control B experimental unit. Excessive TPH levels will interfere with the plant's nutrient metabolism, which will hinder biosynthesis of bioactive metabolites including aloin.

Phenolics, flavonoids and acemannan are essential compounds, mainly responsible for anti-inflammatory, antimicrobial and antioxidant effects; therefore, a reduction in their concentration is worrisome in the pharmaceutical sector [22,46]. The low DPPH scavenging activity noted in the control B extract, is similar to the findings of scholars [18,62], during their clinical investigation into the impact of crude oil spillage, on the antioxidant properties of medical plants. Furthermore, this study's findings show that both the natural and synthetized additives, used as remediating agents, were able to recover these vital phytochemicals within the plant, as demonstrated by the results obtained from the extract analysis. This affirms previous scholars' reports [63,64], which state that natural additives have the ability, to increase the vitamins and bioactive compound concentrations in plants, hence enhancing their nutritional and medical qualities.

Explicitly, the remediating units containing organic materials or biostimulant (T1, T3, T4 and T5) had higher antioxidant activity, TPC, and TFC values. This can be attributed to the higher amounts of organic manures and amino acids, which they contained. This assisted in the formation of essential bioactive compounds, leading to better antioxidant capability of medicinal plants [4]. The TPC and TFC values documented in this research, were greater than those previously documented [46]. This could be attributed to the maturity level of the *Aloe vera*, the extract processing method, the retention time and temperature, the soil fertility level, and the grade of the chemicals used in the experiments. According to the results, the combined treatments aided rapid recovery of the bioactive compounds, typically the depreciation of

antioxidant activity, TPC and TFC caused by environmental pollution. This is a positive development in the medical field. These compounds play essential roles in stabilizing scavenging chemicals (reducing oxidative stress), in the development of nutritional therapy, as well as in the prevention and management of chronic diseases [65,66].

Antioxidants play vital roles in maintaining the stability and integrity of herbal medicines; hence, a reduction in the anti-oxidants level of plant extracts can accelerate rapid degradation of herbal products. This has a lot of consequences on the product integrity and public health safety. Specifically, both the nutritional and medicinal properties of the *Aloe vera* were studied in this investigation, and the findings will help to evaluate if the remediation programs were able to recover the nutraceutical properties of the *Aloe vera* plant. Phytochemical and nutrient components of the treatments tend to aid, the stimulation of secondary metabolites, such as phenolics, flavonoids, and anthraquinones in the plant [70]. Additionally, the improved soil nutrient content caused by the treatment agents, helps to enhance the biosynthesis of antioxidant compounds, thereby increasing the phytochemicals content of the plant's extract.

### Fourier transform infrared analysis (FITR) analysis

The results of the FTIR analysis are displayed in Fig 3 (Plates a to g). Fourier Transform Infrared analysis detects slight variations in the extract's functional groups, which are linked to petroleum contamination. The FTIR diagrams revealed that the remediation improved the extract quality, by reducing the concentration petroleum hydrocarbons in the

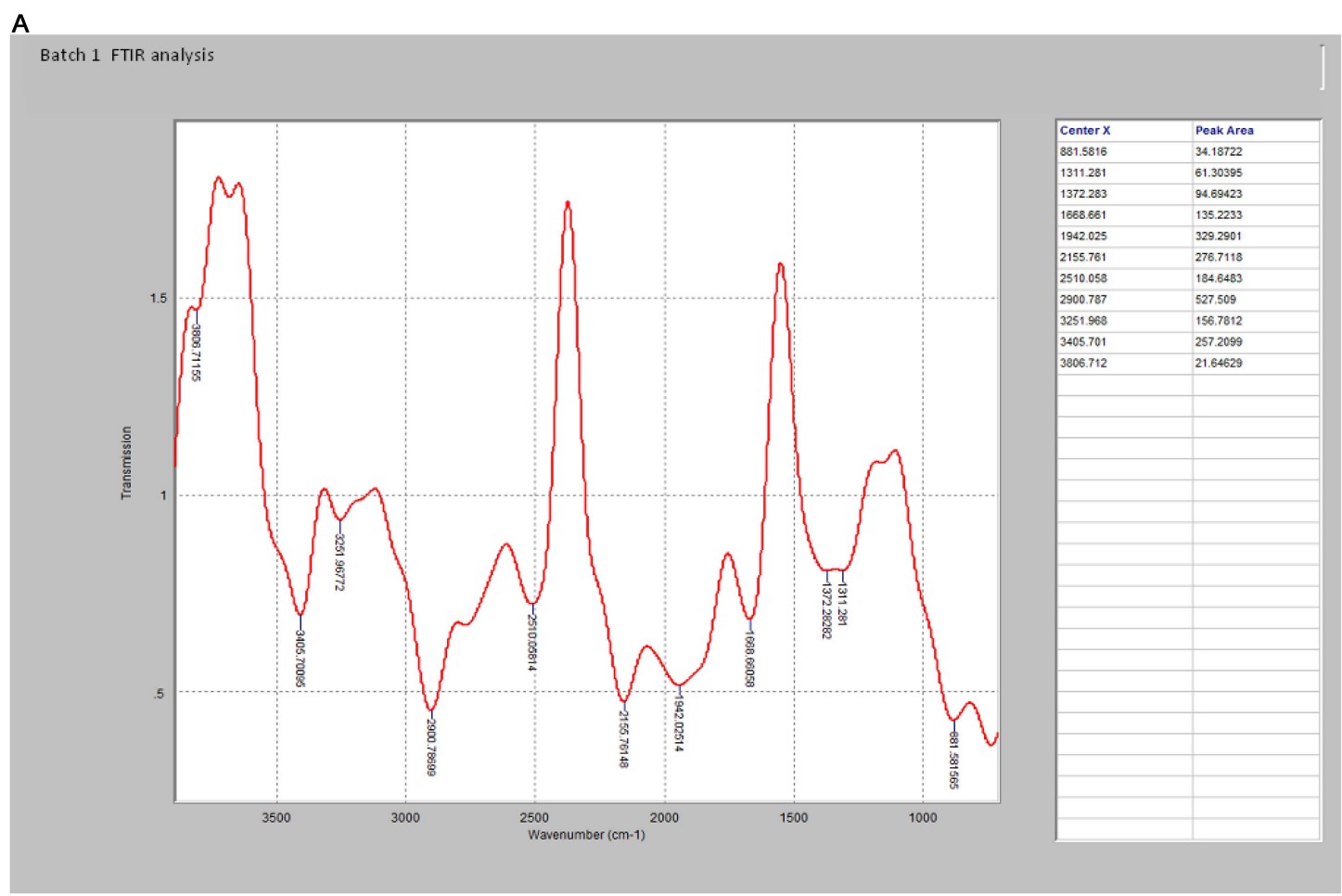

**B**

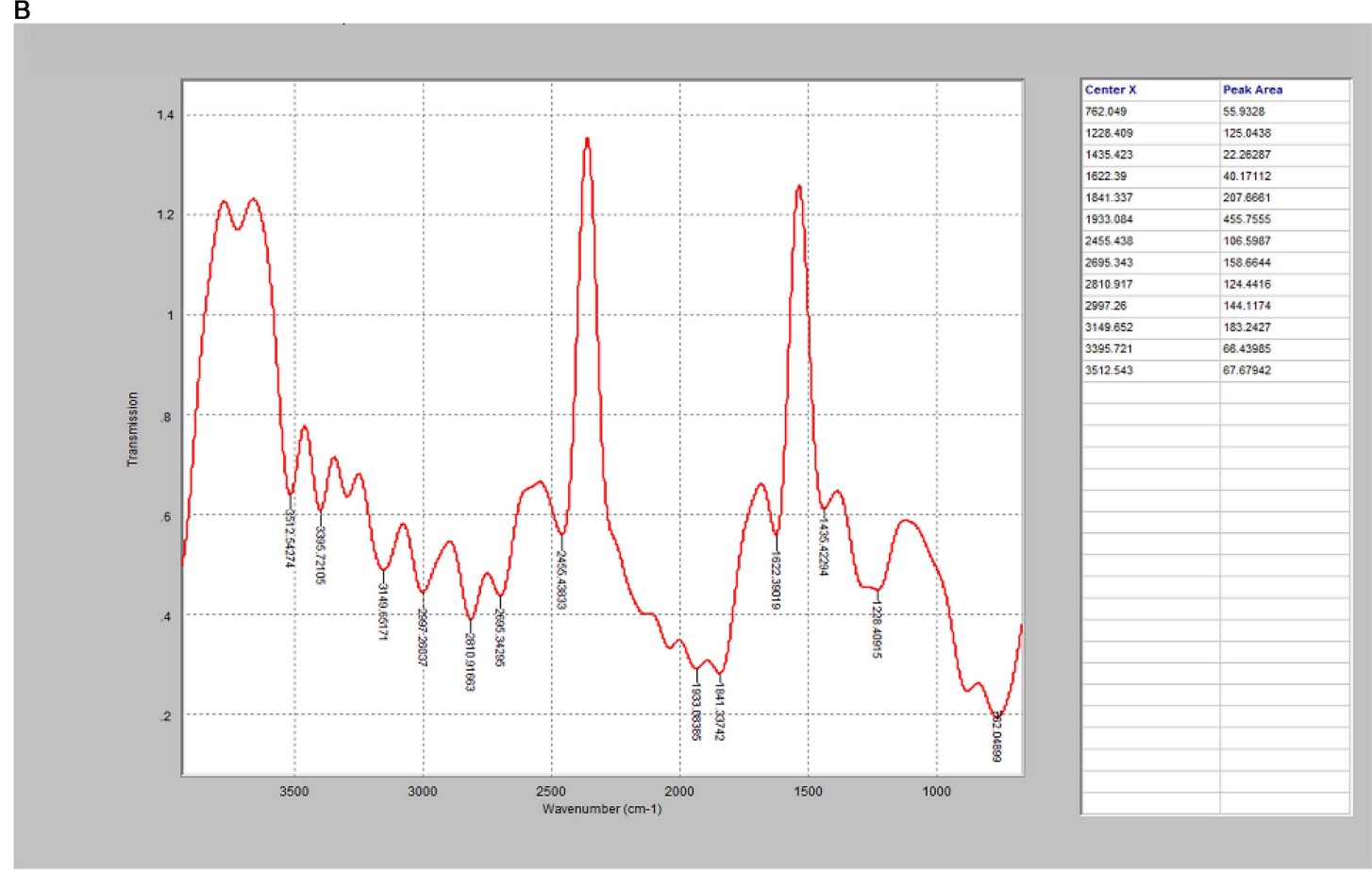

| Center X | Peak Area |
|----------|-----------|
| 762.049 | 55.9328 |
| 1228.409 | 125.0438 |
| 1435.423 | 22.26287 |
| 1622.39 | 40.17112 |
| 1841.337 | 207.6661 |
| 1933.084 | 455.7555 |
| 2455.438 | 106.5987 |
| 2695.343 | 158.6644 |
| 2810.917 | 124.4416 |
| 2997.26 | 144.1174 |
| 3149.652 | 183.2427 |
| 3395.721 | 66.43985 |
| 3512.543 | 67.67942 |

extract. The comparison of the profiles in Plates a to g, portray significant fluctuations in the reduction patterns of the hydrocarbon-related peaks, which indicate the efficacy of the different remediation methods used in the different experimental plans. The FTIR analysis established that the untreated contaminated soil (Control B), exhibited strong C–H peaks, ranging from 2850–2950 $cm^{-1}$, signifying a higher presence of petroleum hydrocarbons. Conversely, the treated soil samples exhibited weaker C–H peaks, which can be attributed to bioremediation actions. Plate a (Control A) does not establish strong confirmation of aromatic hydrocarbons presence in the extract. The polar functional groups were distinct and resilient, which signifies that the plant's juice (fluid) from which the extract was produced, undergone little petroleum hydrocarbons degradation.

The presence of petroleum hydrocarbons inside the plant extract, as detected by FTIR analysis, confirmed their bioavailability in the soil and later absorption by the plant during its growth stage. The FTIR analysis specifically affirmed the occurrence of considerable petroleum hydrocarbons in the control B extract, as well as the extracts from Treatments 1–5 experimental units, as depicted by characteristic absorption bands in the 2950–2850 $cm^{-1}$ regions. Also, the occurrence of prominent hydroxyl, carboxyl, and phenolic groups in an extract is an indication of biodegradation process [67]; hence, it can be seen that T1, T3, T4 and T5 have undergone substantial bioremediation activities during the experimental period (*Aloe vera* growing period). It was observed that the T4 extract (Plate f) exhibited the best remediation efficiency, showing significantly reduced hydrocarbon peaks, which is an evidence of the highest concentrations of bioactive compounds

C

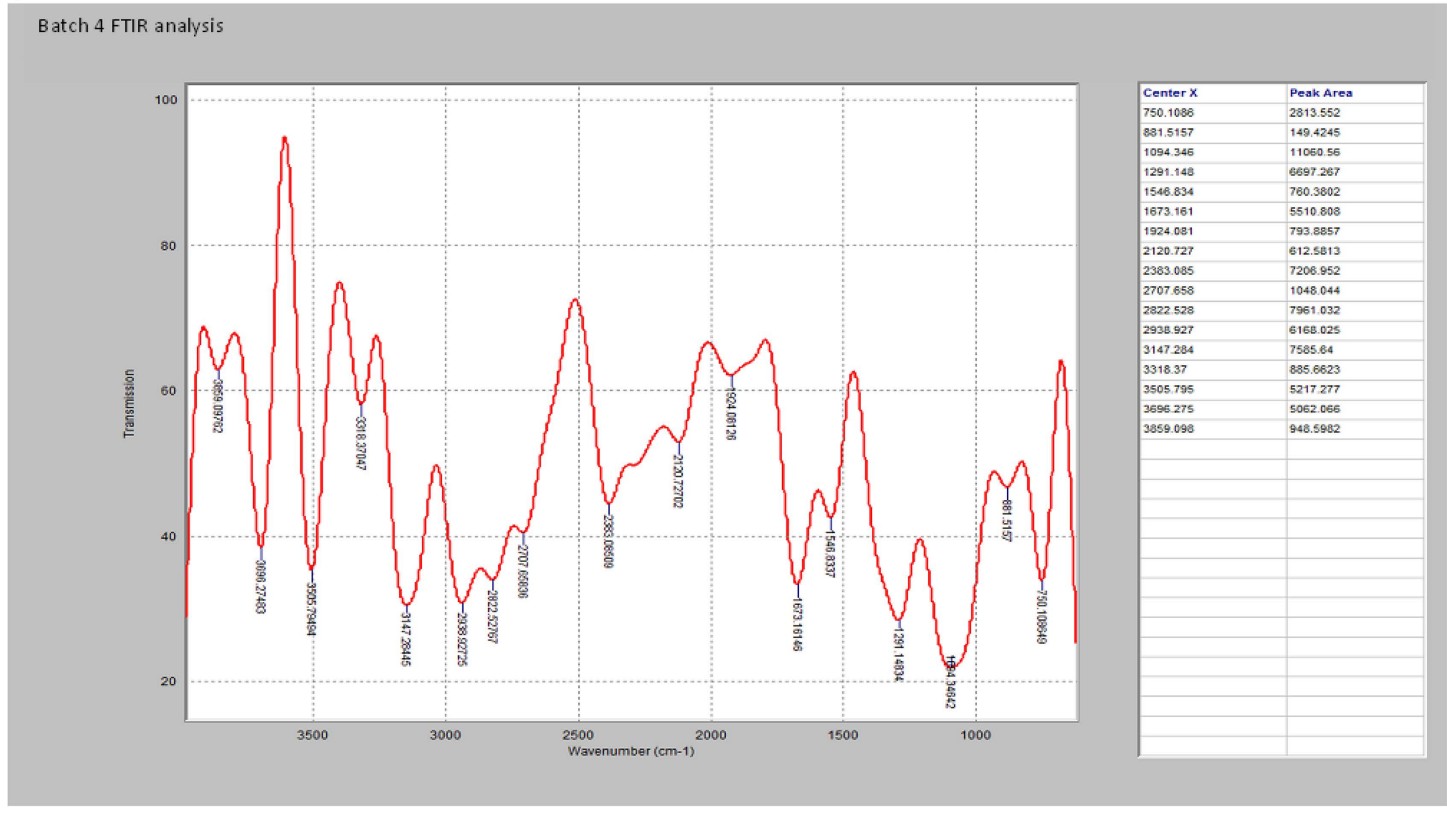

recorded in the T4 extract. Furthermore, the FTIR analysis revealed the synergy between biostimulant (seaweed extract), organic manure, and inorganic fertilizer can considerably increase the degradation of petroleum hydrocarbons in the soil, leading to improved remediation [60]. According to Uguru [16], an effective remediation approach facilitates the rapid degradation of petroleum hydrocarbons, and other toxic substances in the plant system. This leads to healthier plants with improved nutritional quality and enhanced public health safety.

Plate a: The control A extract FTIR diagram
Plate b: The FTIR diagram of control B extract
Plate c: FTIR diagram of T1 extract
Plate d: FTIR diagram of T2 extract
Plate e: FTIR diagram of T3 extract
Plate f: FTIR diagram of T4 extract
Plate h: FTIR diagram of T5 extract

## Pearson correlation

The result of the Pearson correlation analysis of the major parameter, investigated in this study is presented in Table 7. The analysis depicted that the PHs levels have an excellent negative correlation (r ≥ −0.95) with the vitamin A, vitamin C, vitamin E, amino acids, aloin, acemannan, DPPH scavenging activity, TPC, and TFC in the extract. This portrayed that an increase in TPH contamination causes, a significant decline in the parameters concentration of these

D

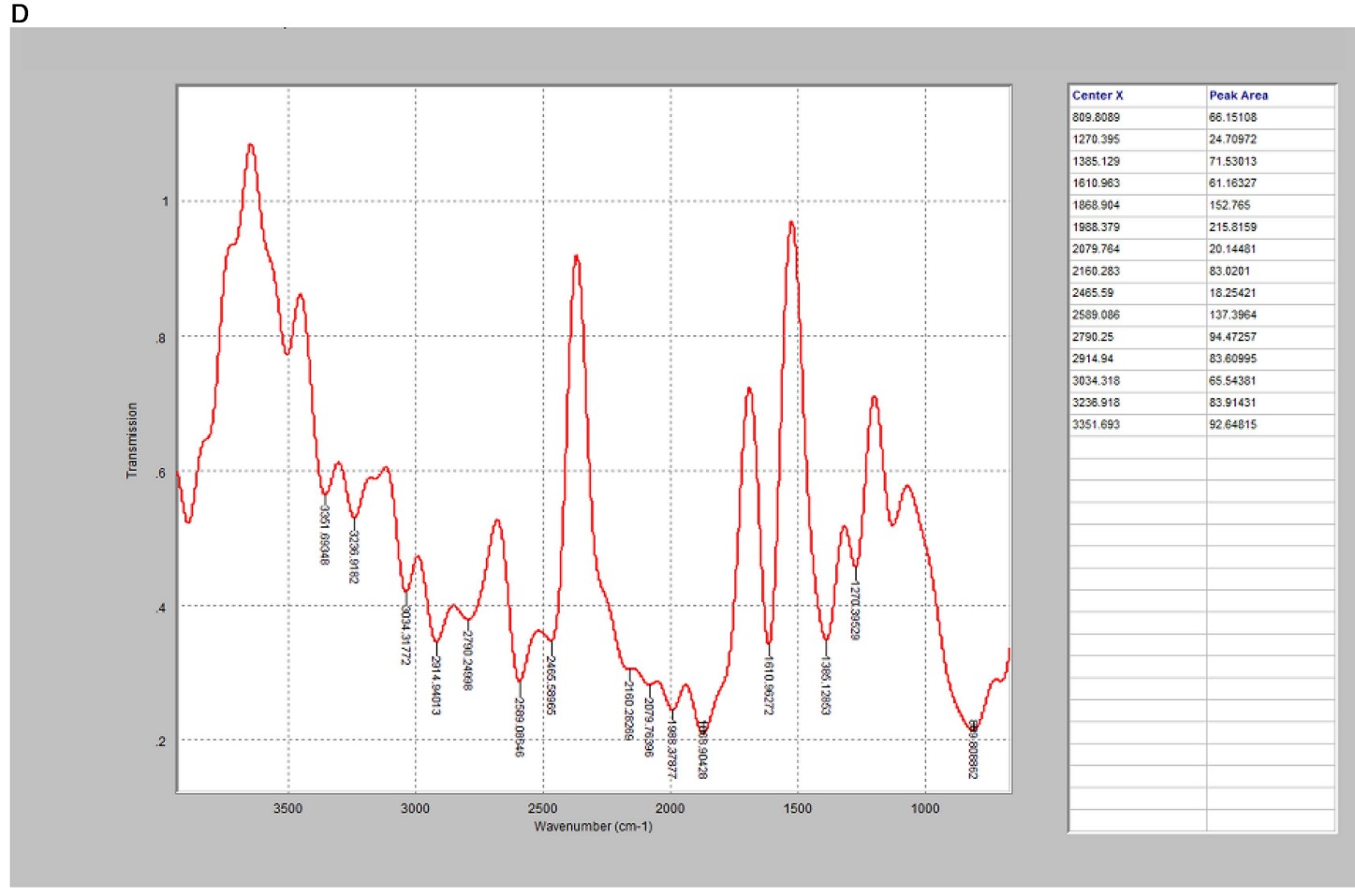

| Center X | Peak Area |
|----------|-----------|
| 809.8089 | 66.15108 |
| 1270.395 | 24.70972 |
| 1385.129 | 71.53013 |
| 1610.963 | 61.16327 |
| 1868.904 | 152.765 |
| 1988.379 | 215.8159 |
| 2079.764 | 20.14481 |
| 2160.283 | 83.0201 |
| 2465.59 | 18.25421 |
| 2589.086 | 137.3964 |
| 2790.25 | 94.47257 |
| 2914.94 | 83.60995 |
| 3034.318 | 65.54381 |
| 3236.918 | 83.91431 |
| 3351.693 | 92.64815 |

parameters. Also, the Pearson analysis further shows that a strong positive correlation existed, between the phyto-chemicals with the DPPH levels of the extract. This is an indication that, the phytochemicals substantially enhance the extract DPPH inhibition activity. Specifically, the analysis has highlighted that the petroleum contamination, has a substantial impact in the phytochemicals synthesis, resulting in poorer antioxidant activity. This is in conformity with earlier observation on the extract, which was produced from the leaves of *Hyptis suaveolens* (L.) cultivated in crude oil contaminated area [18].

### Practical implications of the study and future research

The data provided by this study have established that the hybridization of bio-materials such as *Ascophyllum nodosum*, rice husk, cow dung, poultry waste, kitchen waste, and oil palm empty fruit bunches, can effectively help degrade the concentration of petroleum hydrocarbons in the soil. This leads to a partial restoration of the phytochemicals' concentration in herbal plants. Also, the findings have affirmed that eco-friendly treatment therapies can be successfully used, to increase the concentration of essential bioactive compounds in herbal plants, resulting in their broader applications in complementary medicines. Remarkably, this study's experimental framework (utilization of eco-friendly materials) has a wide range of environmental remediation applications. Aside from petroleum hydrocarbons degradation, these non-chemical materials

E

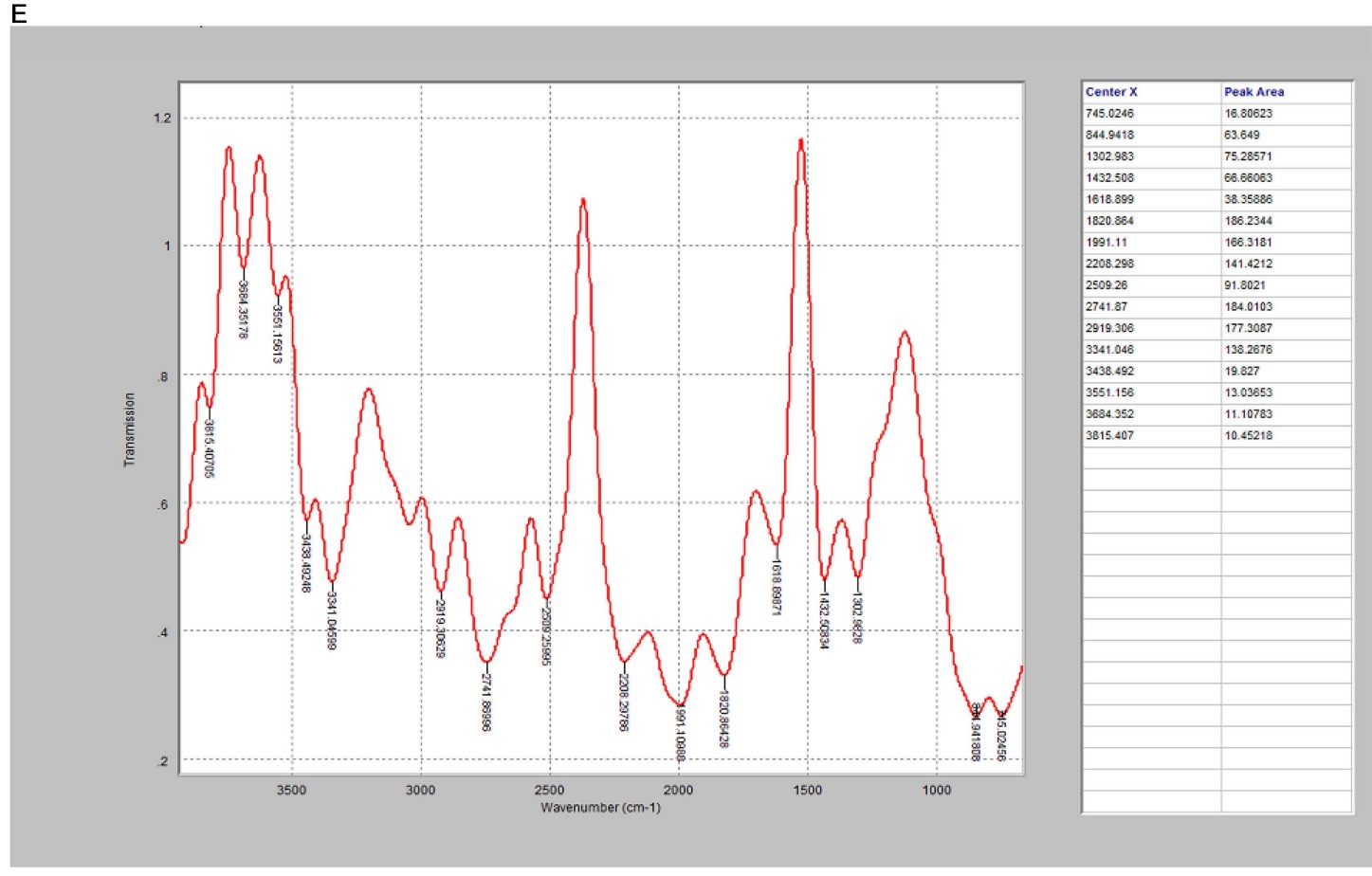

can be used to remediate contaminated media (farm lands), where agricultural and medicinal plants are grown. However, the results of this study were obtained through pot experiments and not field trials; therefore, on-site (*in situ*) confirmations are recommended before the commercialization of the study's findings.

Particularly, this study's findings make numerous contributions to public health safety. The bio-amendments, as affirmed by the FTIR analysis, were able to significantly reduce the toxicity associated with petroleum contamination, and enhance the pharmacological efficacy and quality of the plant extract. As presented by previous reports, higher petroleum concentration (greater than 20%) in the soil will lead to higher TPH levels, which will have severe negative implications on the herbal plants [68,69]. These conditions will reduce the treatment's remediation efficiency, and phytochemical qualities of the plant; on the contrary, higher treatment concentrations will be required to attain a better antioxidant restoration level. Therefore, the only concentration level used in this study is a major limitation; hence, further experiments involving a wide range of petroleum concentrations should be conducted.

## Conclusion

*Aloe vera* plant has several vital bioactive compounds, which include phenolics, flavonoids, and vitamins that have potent antioxidant, anti-inflammatory, and antimicrobial properties. However, environmental pollution such as petroleum pollution tends to have a significant destructive impact on the *Aloe vera*'s essential phytochemicals. This scientific investigation

F

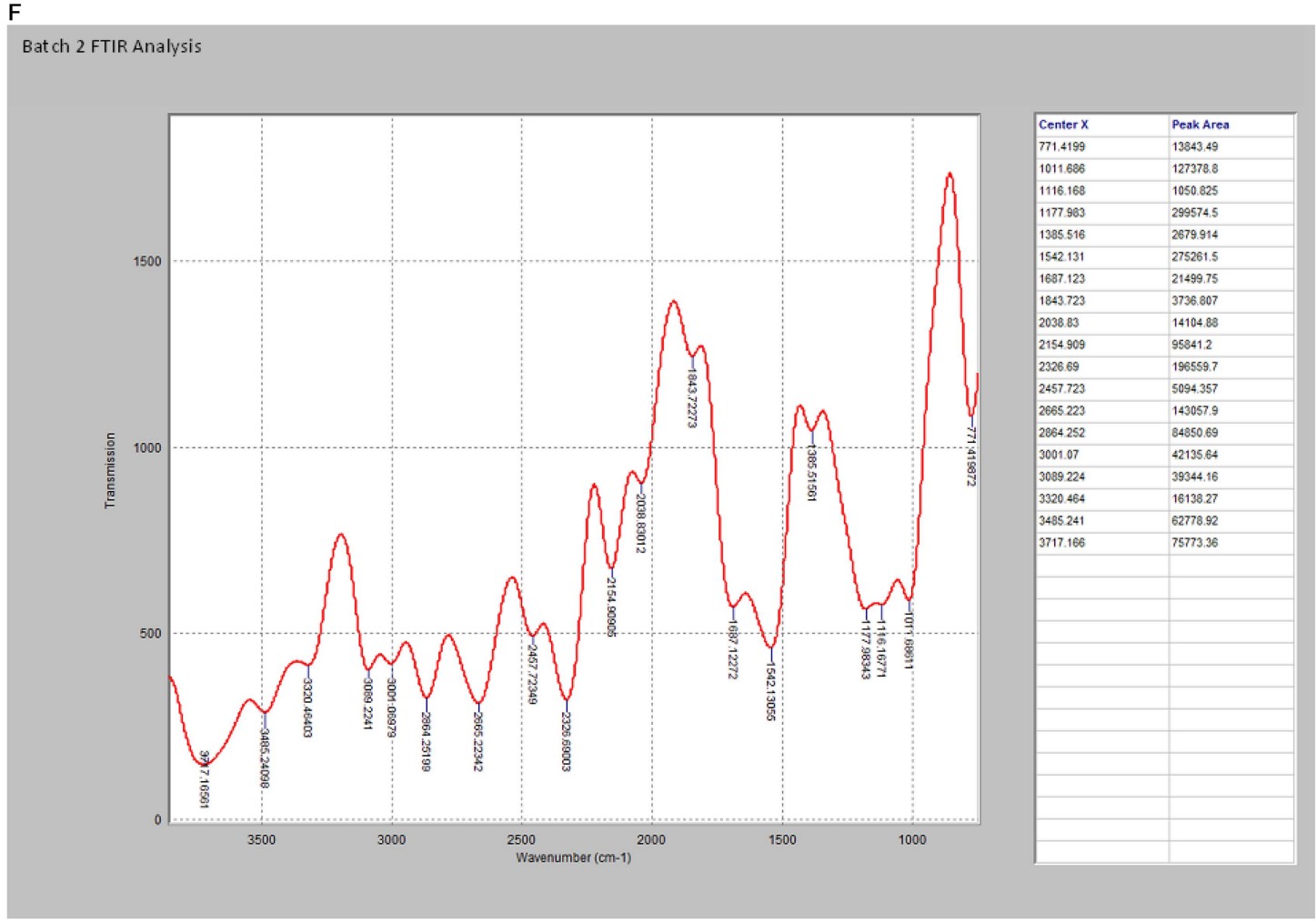

Batch 2 FTIR Analysis

| Center X | Peak Area |
|---|---|
| 771.4199 | 13843.49 |
| 1011.686 | 127378.8 |
| 1116.168 | 1050.825 |
| 1177.983 | 299574.5 |
| 1385.516 | 2679.914 |
| 1542.131 | 275261.5 |
| 1687.123 | 21499.75 |
| 1843.723 | 3736.807 |
| 2038.83 | 14104.88 |
| 2154.909 | 95841.2 |
| 2326.69 | 196559.7 |
| 2457.723 | 5094.357 |
| 2665.223 | 143057.9 |
| 2864.252 | 84850.69 |
| 3001.07 | 42135.64 |
| 3089.224 | 39344.16 |
| 3320.464 | 16138.27 |
| 3485.241 | 62778.92 |
| 3717.166 | 75773.36 |

was conducted to appraise the impact of petroleum contamination, and different remediation approaches on the production, pharmacologically active compounds, and antioxidant activity of *Aloe vera* extract. The results of the laboratory tests revealed that, the petroleum contamination has a substantial influence on the plant's nutritional and medical properties; however, the soil remediation strategies successfully alleviate these adverse effects. The FTIR and Pearson correlation analyses, further confirmed that a perfect negative correlation occurred between the petroleum pollution, and the bioactive compounds concentration in the *Aloe vera* extract. Particularly, the treatments caused a substantial decline in soil and extract TPH levels, as well as, significant recoveries of phytochemicals and antioxidant activity in the plant extract. This restored the phytochemicals and antioxidant integrity of the *Aloe vera* plant. It was observed that the T4 hybridized treatment exhibited the best results, with the *Aloe vera* extract displaying the lowest TPH value, and the highest vitamin and amino-acid recovery rates. Compared to the Control B results, it was noted that the vitamin A, vitamin C, vitamin E, total amino acids, aloin, acemannan, DPPH, total phenolics, and total flavonoids levels, in the T4 extract increased by 94.5, 118.1, 66.4, 237.7, 87.0, 92.2, 182.1, 132.0, and 235.9%, respectively. These data have revealed that these treatments are perfect, eco-friendly solutions with potent bioremediation effectiveness, relieving pollution stress in plants, and restoring the bioactive compounds of plants growing in petroleum-contaminated environments. A major limitation of this

G

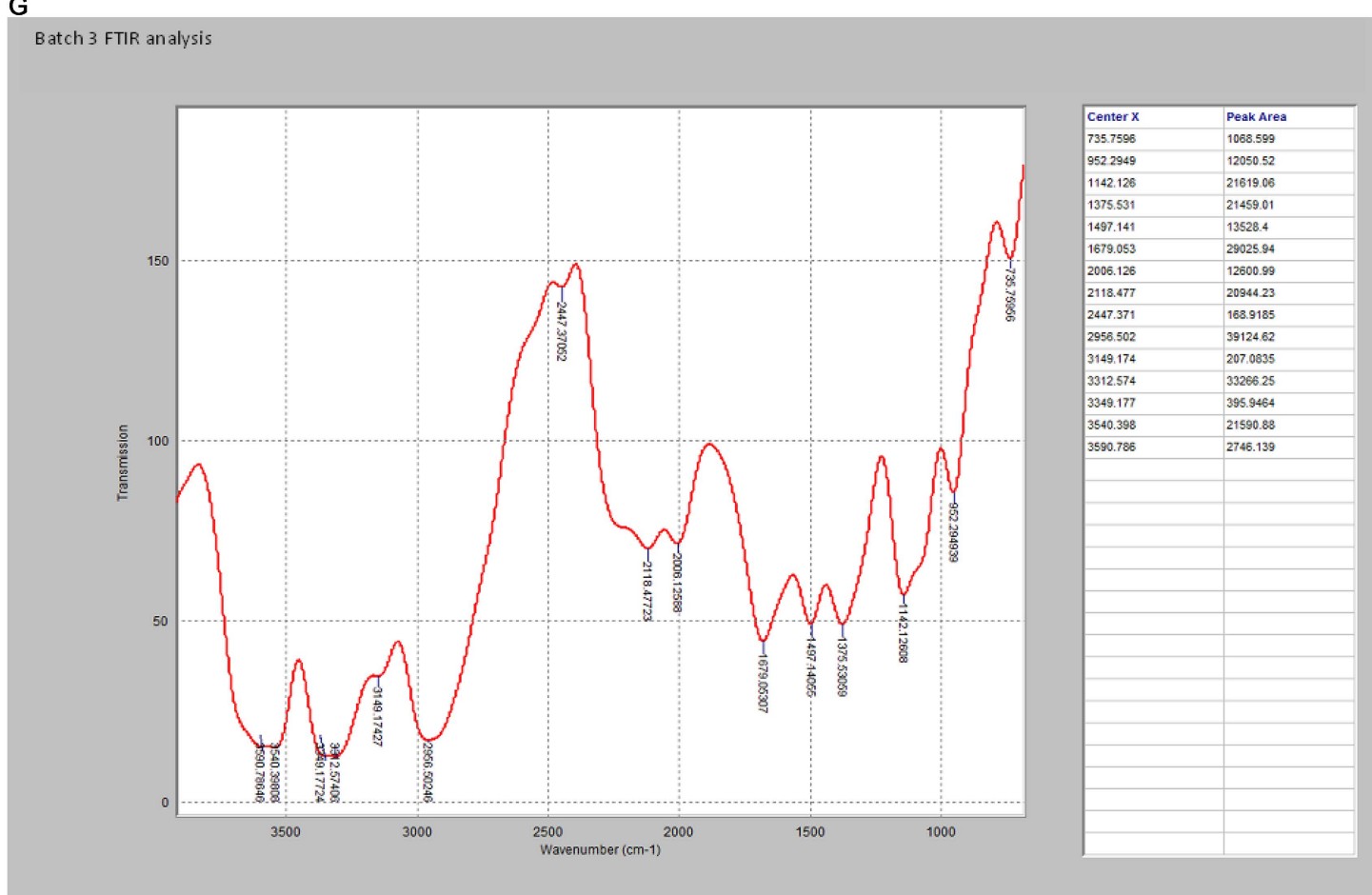

**Fig 3. The FTIR profiles of the various extracts.**

**Table 7. The Pearson correlation analysis.**

|  | Alio | Acw | DPPH | Vit A | Vit C | Vit E | AC | ETPH | TPC | TFC |
|---|---|---|---|---|---|---|---|---|---|---|
| **Alio** | 1 | | | | | | | | | |
| **Acw** | 0.989** | 1 | | | | | | | | |
| **DPPH** | 0.977** | 0.979** | 1 | | | | | | | |
| **Vit A** | 0.968** | 0.959** | 0.973** | 1 | | | | | | |
| **Vit C** | 0.990** | 0.979** | 0.987** | 0.983** | 1 | | | | | |
| **Vit E** | 0.965** | 0.958** | 0.948** | 0.962** | 0.958** | 1 | | | | |
| **AC** | 0.985** | 0.976** | 0.972** | 0.945** | 0.975** | 0.946** | 1 | | | |
| **ETPH** | 0−.982** | −0.958** | −.0951** | −0.955** | −0.976** | −0.963** | −0.967** | 1 | | |
| **TPC** | 0.991** | 0.973** | 0.962** | .0955** | 0.975** | 0.959** | 0.986** | −0.985** | 1 | |
| **TFC** | 0.972** | 0.967** | 0.974** | 0.982** | 0.979** | 0.947** | 0.963** | −0.955** | .0968** | 1 |

` ** Significant at 0.01 level (2-tailed), Vit A – vitamin A, Vit C – vitamin C, Vit E – vitamin E, AC – amino acid, Alio – aloin, Acw – acemannan, ETPH – extract TPH.

study is that, it is a pot study (not a field experiment), involving only one petroleum contamination level. Additionally, due to a lack of funds, only five treatments were evaluated in this study. Therefore, the results are not robust enough. Future studies involving more contamination levels, additional clinical tests, and focusing on field trials are required, to produce more robust results.

## Supporting information

**S1 File. Vitamins HPLC profile.**
(DOCX)

## Acknowledgments

The authors would like to acknowledge the Deanship of Graduate Studies and Scientific Research, Taif University for funding this work.

## Author contributions

**Conceptualization:** Sarah Alharthi, Idisi Benjamin Evi, Sara M. Almutairi.

**Data curation:** Sarah Alharthi, Uguru Hilary, Idisi Benjamin Evi, Mahmoud Helal.

**Formal analysis:** Sarah Alharthi, Amal A. Alyamani, Rokayya Sami, Mahmoud Helal.

**Funding acquisition:** Sarah Alharthi, Amal A. Alyamani, Rokayya Sami, Uguru Hilary, Mahmoud Helal, Salma M. Aljahdali, Afnan M. Alnajeebi.

**Investigation:** Rokayya Sami, Uguru Hilary, Idisi Benjamin Evi, Akpokodje Ovie Isaac, Salma M. Aljahdali, Moayad M. Khashoqji, Afnan M. Alnajeebi, Hayat A. Alghamdi.

**Methodology:** Sarah Alharthi, Ola A. Abu Ali, Amal A. Alyamani, Nashi K. Alqahtani, Rokayya Sami, Uguru Hilary, Idisi Benjamin Evi, Akpokodje Ovie Isaac, Haneen H. Mouminah, Norah E. Aljohani, Mahmoud Helal, Salma M. Aljahdali, Moayad M. Khashoqji, Afnan M. Alnajeebi, Hayat A. Alghamdi, Sara M. Almutairi.

**Project administration:** Nashi K. Alqahtani, Mahmoud Helal, Moayad M. Khashoqji, Afnan M. Alnajeebi.

**Resources:** Amal A. Alyamani, Moayad M. Khashoqji.

**Software:** Amal A. Alyamani, Nashi K. Alqahtani, Rokayya Sami, Akpokodje Ovie Isaac, Salma M. Aljahdali, Moayad M. Khashoqji, Sara M. Almutairi.

**Supervision:** Akpokodje Ovie Isaac, Haneen H. Mouminah, Norah E. Aljohani, Hayat A. Alghamdi, Sara M. Almutairi.

**Validation:** Ola A. Abu Ali, Rokayya Sami, Norah E. Aljohani, Sara M. Almutairi.

**Visualization:** Ola A. Abu Ali, Haneen H. Mouminah, Norah E. Aljohani, Afnan M. Alnajeebi.

**Writing – original draft:** Ola A. Abu Ali, Nashi K. Alqahtani, Rokayya Sami, Haneen H. Mouminah, Norah E. Aljohani.

**Writing – review & editing:** Ola A. Abu Ali, Rokayya Sami, Haneen H. Mouminah, Sara M. Almutairi.

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
