## [Decision Letter · Decision Letter 0]

3 Dec 2025

PONE-D-25-61186

HPLC and FTIR Analysis of Phytochemicals and Antioxidants of Aloe vera Exposed to Petroleum Hydrocarbons and Remediation Treatments with Organic and Inorganic Amendments

PLOS ONE

Dear Dr. Sami,

Thank you for submitting your manuscript to PLOS ONE. After careful consideration, we feel that it has merit but does not fully meet PLOS ONE’s publication criteria as it currently stands. Therefore, we invite you to submit a revised version of the manuscript that addresses the points raised during the review process.

We look forward to receiving your revised manuscript.

Kind regards,

Nishant Kumar, Ph.D

Academic Editor

PLOS ONE

Journal Requirements:

“Acknowledgments

The authors extend their appreciation to Taif University, Saudi Arabia, for supporting this work through project number (TU-DSPP-2024-79).

Funding

This research was funded by Taif University, Saudi Arabia, Project No. (TU-DSPP-2024-79).”

6. Please amend the manuscript submission data (via Edit Submission) to include author Sarah Alharthi, Ola Abu Ali and Amal A. Alyamani.

Reviewers' comments:

Reviewer's Responses to Questions

**Comments to the Author**

1. Is the manuscript technically sound, and do the data support the conclusions?

Reviewer #1: Yes

Reviewer #2: Yes

Reviewer #3: Yes

2. Has the statistical analysis been performed appropriately and rigorously? 

Reviewer #1: Yes

Reviewer #2: Yes

Reviewer #3: Yes

3. Have the authors made all data underlying the findings in their manuscript fully available?

Reviewer #1: Yes

Reviewer #2: Yes

Reviewer #3: Yes

4. Is the manuscript presented in an intelligible fashion and written in standard English?

Reviewer #1: Yes

Reviewer #2: Yes

Reviewer #3: Yes

5. Review Comments to the Author

Reviewer #1: Comments for Author

Dear Authors,

This manuscript entitled: “ HPLC and FTIR Analysis of Phytochemicals and Antioxidants of Aloe vera Exposed to Petroleum Hydrocarbons and Remediation Treatments with Organic and Inorganic Amendments’’

Is important for the scientific community, through which a species of medicinal plants (Aloe vera) belonging to the Asphodelaceae family was described, mentioning the characteristics of this type.

The work is interesting but it needs to be improved by answering the previous questions.

Please answer those questions:

1. What were the exact compositions and application rates of the organic manure, potassium permanganate, and seaweed extract in each treatment (T1–T5)?

2. How were Aloe vera seedlings selected, standardized, and transplanted into the experimental units?

3. What environmental conditions (light, temperature, watering schedule) were maintained during the 12-week growth period?

4. How were soil samples collected and processed to determine total petroleum hydrocarbons (TPH)?

5. What protocol was used to extract acemannan and aloin before HPLC analysis?

6. What reagents and procedures were used to determine total phenolic content (TPC)?

7. How were antioxidant activities assessed (e.g., DPPH, ABTS, or FRAP method)?

8. What procedure was used to analyze amino acid composition in the Aloe vera extract?

9. How were the recovery rates calculated for the remediation treatments?

10. How were statistical analyses performed to determine significant differences among treatments?

Reviewer #2: 1. How strong is the evidence that eco-friendly materials are effective for remediating petroleum-contaminated soils?

2. What specific eco-friendly treatment component contributed most to the successful outcome?

3. What are the practical implications for farmers cultivating medicinal plants in polluted areas?

4. How do these findings contribute to public safety and medicinal plant quality assurance?

5. How might the results differ if contamination levels were higher or lower than 20% crude oil?

6. What long-term effects on soil health are implied by the successful treatment?

7. Do the authors recommend any further research to validate the remediation strategy?

8. How might these findings influence environmental policies regarding petroleum-polluted farmlands?

9. What limitations did the authors acknowledge regarding the results or methods?

10. How do the findings compare with previous studies on petroleum contamination in medicinal plants?

11. How do these conclusions support the use of eco-friendly materials over chemical remediation methods?

Reviewer #3: HPLC and FTIR Analysis of Phytochemicals and Antioxidants of Aloe vera Exposed to Petroleum Hydrocarbons and Remediation Treatments with Organic and Inorganic Amendments".

Report

The subject is very interesting for application Aloe vera. The manuscript can be accepted for the publication in ploes after addressing the following comments. Below I present the observations related to the content.

1. How did the concentrations of vitamins A, C, and E differ between the contaminated control and the best-performing treatment (T4)?

2. What specific trends were observed in total phenolic content (TPC) across the five treatments?

3. How did petroleum contamination affect the total flavonoid content (TFC) compared to uncontaminated soil?

4. Which treatment showed the lowest antioxidant activity, and what might have caused this reduction?

5. Was there a correlation between TPH reduction and recovery of phytochemicals in Aloe vera?

6. How did the presence of petroleum hydrocarbons detected by FTIR differ before and after remediation?

7. Did amino acid composition show any significant changes between treatments?

8. How large was the percentage improvement in nutrient and phytochemical content in treatment T4 compared to the contaminated control?

9. Did any treatment fully restore Aloe vera phytochemicals to levels similar to uncontaminated soil?

10. How did each treatment affect the antioxidant capacity of the Aloe vera extract?

11. Was there a significant relationship between vitamin concentrations and the soil remediation method used?

12. Were any of the VOC-related or aroma-related properties (if measured) affected by petroleum contamination?

6. PLOS authors have the option to publish the peer review history of their article (what does this mean? ). If published, this will include your full peer review and any attached files.

**Do you want your identity to be public for this peer review?** For information about this choice, including consent withdrawal, please see our Privacy Policy .

Reviewer #1: **Yes:** BENABDERRAHMANE Wassila

Reviewer #2: No

Reviewer #3: No

---

## [Author Response · Author response to Decision Letter 1]

17 Dec 2025

Response

PONE-D-25-61186

HPLC and FTIR Analysis of Phytochemicals and Antioxidants of Aloe vera Exposed to Petroleum Hydrocarbons and Remediation Treatments with Organic and Inorganic Amendments

PLOS ONE

Reviewer's Responses to Questions

Reviewer #1: Comments for Author

Comment 1: What were the exact compositions and application rates of the organic manure, potassium permanganate, and seaweed extract in each treatment (T1–T5)?

Answer:the details are explained in lines 165 to 172

Comment 2: How were Aloe vera seedlings selected, standardized, and transplanted into the experimental units?

Answer:Plants were collected from the Department of Agricultural Engineering research center (15 July 2024) and authenticated by a crop scientist (22 July 2024). Check line 141 to 144

Aloe vera was standardized by planting it in uncontaminated loamy soil for three weeks, during which tap water was provided to the plantlets whenever the soil moisture content fell below 40% (wet basis). Check lines 201 to 203

The three-week-old Aloe vera plantlets were manually inspected, and only strong, vigorous, and disease-free plantlets were transplanted into the experimental pots. Check lines 205 to 206

Comment 3. What environmental conditions (light, temperature, watering schedule) were maintained during the 12-week growth period?

Answer:Sunlight: ~10 hours/day; Temperature: 24–37 °C; Relative Humidity: 81–95%; Wind Speed: 3.2–12.1 km/h. Check line 212 to 215

Comment 4: How were soil samples collected and processed to determine total petroleum hydrocarbons (TPH)?

Answer:Check line 249 to 251, 260 - 262

Comment 5. What protocol was used to extract acemannan and aloin before HPLC analysis?

Answer:Check line 349 – 350. The complete protocol are presented in Table 1

Comment 6:What reagents and procedures were used to determine total phenolic content (TPC)?

Answer:Check line 281 - 285

Comment 7:How were antioxidant activities assessed (e.g., DPPH, ABTS, or FRAP method)?

Answer:Check line 384 - 386

Comment 8: What procedure was used to analyze amino acid composition in the Aloe vera extract?

Answer:Check line 370 - 373

Comment 9: How were the recovery rates calculated for the remediation treatments?

Answer:Check line 394– 397

Recovery rate (%)= (specific treatment value)/(Control treatment value)×100

Comment 10: How were statistical analyses performed to determine significant differences among treatments?

Answer:Check line 400 – 405. One-way analysis of variance (ANOVA

Reviewer #2:

Comment 1: How strong is the evidence that eco-friendly materials are effective for remediating petroleum-contaminated soils?

Answer:Corrected; check the conclusion

Comment 2: What specific eco-friendly treatment component contributed most to the successful outcome?

Answer:Corrected; check the abstract lines 60 to 63

Comment3: What are the practical implications for farmers cultivating medicinal plants in polluted areas?

Answer:Corrected: it has been discussed under the “practical implications of the study and future research” sub section, check lines 735 to 760

Comment 4. How do these findings contribute to public safety and medicinal plant quality assurance?

Answer:Corrected: it has been discussed under the “practical implications of the study and future research” sub section, check lines 735 to 738

Comment 5: How might the results differ if contamination levels were higher or lower than 20% crude oil?

Answer:Corrected: it has been discussed under the “practical implications of the study and future research” sub section, check lines 735 to 744

Comment 6:What long-term effects on soil health are implied by the successful treatment?

Answer:Discussed: check lines 483 to 486

Comment 7: Do the authors recommend any further research to validate the remediation strategy?

Answer:Yes: --- evaluation of pharmacokinetics and therapeutic effects of Aloe vera--- check lines 774 to 776

Comment 8:How might these findings influence environmental policies regarding petroleum-polluted farmlands?

Answer::Corrected: it has been discussed under the “practical implications of the study and future research” sub section, check lines 743 to 746

Comment 9: What limitations did the authors acknowledge regarding the results or methods?

Answer:Yes: check lines 785 to 790

Comment 10:. How do the findings compare with previous studies on petroleum contamination in medicinal plants?

Answer:The study aligns with earlier reports that petroleum contamination reduces vitamins, phenolics, flavonoids and antioxidant activity in medicinal plants, more than 20 related literatures were cited and compared

Comment 11:. How do these conclusions support the use of eco-friendly materials over chemical remediation methods?

Answer:Compared to the inorganic (KMnO₄) treatment or KMnO₄-based hybrids, organic and biostimulant treatments (OM, ISE, and particularly their combination T4) resulted in higher TPH reduction in soil and extract as well as better recovery of vitamins, amino acids, phytochemicals, and antioxidant capacity. Refer to Tables 4 and 5–6.

Reviewer #3:

Comment 1: How did the concentrations of vitamins A, C, and E differ between the contaminated control and the best-performing treatment (T4)?

Answer:Specifically, the vitamins A, C, and E concentrations in the Aloe vera planted in contaminated soil increased by 48.58%, 54.15%, and 39.89%, respectively, after T4 application. Check lines 499 to 504

Comment 2: What specific trends were observed in total phenolic content (TPC) across the five treatments?

Answer:The TPC and TFC levels in the extract declined sharply (61.58% and 74.99% respectively) after petroleum contamination. These parameters (TPC and TFC) increased gradually after the soil amendments, with T4 sample attaining the maximum TPC and TFC recovery rates (89.22% and 84.03%, respectively) among the five treatment units.

Check lines 595 to 603

Comment 3:How did petroleum contamination affect the total flavonoid content (TFC) compared to uncontaminated soil?

Answer:The TPC and TFC levels in the extract declined sharply (61.58% and 74.99% respectively) after petroleum contamination. These parameters (TPC and TFC) increased gradually after the soil amendments, with T4 sample attaining the maximum TPC and TFC recovery rates (89.22% and 84.03%, respectively) among the five treatment units.

Check lines 595 to 603

Comment 4: Which treatment showed the lowest antioxidant activity, and what might have caused this reduction?

Answer:Check lines 600 to 602

Comment 5: Was there a correlation between TPH reduction and recovery of phytochemicals in Aloe vera?

Answer:Yes. Pearson results show a very strong negative correlation between extract TPH and phytochemical levels (r ≈ −0.95 to −0.98)Check lines 716 to 718

Comment 6:. How did the presence of petroleum hydrocarbons detected by FTIR differ before and after remediation?

Answer:It has been explained, check lines 668 to 672

Comment 7:. Did amino acid composition show any significant changes between treatments?

Answer:Yes. amino-acid levels in the extract differed significantly across units. T4 gave the closest recovery (≈92.9% of Control A) but was still statistically different from the uncontaminated control. Check Table 5

Comment 8:. How large was the percentage improvement in nutrient and phytochemical content in treatment T4 compared to the contaminated control?

Answer:Compared to the Control B results, the vitamin A, vitamin C, vitamin E, total amino acids, aloin, acemannan, DPPH, total phenolics, and total flavonoids levels in the T4 extract increased by 94.5%, 118.1%, 66.4%, 237.7%, 87.0%, 92.2%, 182.1%, 132.0%, and 235.9%, respectively

Kindly check lines 778 to 782

Comment 9:Did any treatment fully restore Aloe vera phytochemicals to levels similar to uncontaminated soil?

Answer:No treatment completely restored phytochemical and nutrient levels to the uncontaminated control (Control A). Though Treatment 4 tried a lot

Comment 10:. How did each treatment affect the antioxidant capacity of the Aloe vera extract?

Answer:According to Table 6, Control A (uncontaminated) has the highest antioxidant metric, while T4 had the nest remediation effect

Comment 11:Was there a significant relationship between vitamin concentrations and the soil remediation method used?

Answer:Yes — the Pearson correlation matrix shows very strong positive correlations between vitamin concentrations (A, C, E) and the phytochemical/antioxidant parameters,

Check Table 7,

Comment 12: Were any of the VOC-related or aroma-related properties (if measured) affected by petroleum contamination?

Answer:The study did not present a targeted VOC, we are sorry for this limitation

Extra comments

Comment A1: Delete notably

Answer: corrected

Comment A2: Use the jourbal ref style

Answer: corrected

Comment A3: provide mechanistic explanation showing how pollutants affect plant biochemical pathway

Answer: corrected check lines 79 - 86

Comment A4: more refs needed

Answer: two more recent refs added

Comment A5: break into two smaller sentences

Answer: corrected

Comment A6: Sentence long

Answer: corrected, break into three sentences, check lines 94 to 100

Comment A7: Why do you choose this depth?

Answer: corrected, check lines 148 to 154

Comment A8: Is it 42 or 49

Answer:corrected to 49, check lines 207

Comment A9: Check the grammar

Answer: corrected,

Comment A10: Recast

Answer: corrected,

Comment A11: Check the grammar

Answer: corrected, check lines 348 to 351

Comment A12: Why different color?

Answer: corrected,

Best regards

---

## [Decision Letter · Decision Letter 1]

26 Dec 2025

HPLC and FTIR Analysis of Phytochemicals and Antioxidants of Aloe vera Exposed to Petroleum Hydrocarbons and Remediation Treatments with Organic and Inorganic Amendments

PONE-D-25-61186R1

Dear Dr. Sami,

We’re pleased to inform you that your manuscript has been judged scientifically suitable for publication and will be formally accepted for publication once it meets all outstanding technical requirements.

Kind regards,

Nishant Kumar, Ph.D

Academic Editor

PLOS One

Additional Editor Comments (optional):

-

Reviewers' comments:

Reviewer's Responses to Questions

**Comments to the Author**

1. If the authors have adequately addressed your comments raised in a previous round of review and you feel that this manuscript is now acceptable for publication, you may indicate that here to bypass the “Comments to the Author” section, enter your conflict of interest statement in the “Confidential to Editor” section, and submit your "Accept" recommendation.

Reviewer #1: All comments have been addressed

Reviewer #2: All comments have been addressed

2. Is the manuscript technically sound, and do the data support the conclusions?

Reviewer #1: Yes

Reviewer #2: Yes

3. Has the statistical analysis been performed appropriately and rigorously? 

Reviewer #1: Yes

Reviewer #2: Yes

4. Have the authors made all data underlying the findings in their manuscript fully available?

Reviewer #1: Yes

Reviewer #2: Yes

5. Is the manuscript presented in an intelligible fashion and written in standard English?

Reviewer #1: (No Response)

Reviewer #2: Yes

6. Review Comments to the Author

Reviewer #1: (No Response)

Reviewer #2: The revised version of the manuscript is appropriate and meet the standards set by the journal.The authors did address all the comments adequately.

7. PLOS authors have the option to publish the peer review history of their article (what does this mean? ). If published, this will include your full peer review and any attached files.

**Do you want your identity to be public for this peer review?** For information about this choice, including consent withdrawal, please see our Privacy Policy .

Reviewer #1: **Yes:** Benabderrahmane Wassila

Reviewer #2: No

---

## [Editor Report · Acceptance letter]

PONE-D-25-61186R1

PLOS One

Dear Dr. Sami,

I'm pleased to inform you that your manuscript has been deemed suitable for publication in PLOS One. Congratulations! Your manuscript is now being handed over to our production team.

Kind regards,

on behalf of

Dr. Nishant Kumar

Academic Editor

PLOS One